# Combinatorial chromatin dynamics foster accurate cardiopharyngeal fate choices

Claudia Racioppi*, Keira A Wiechecki, Lionel Christiaen*

Center for Developmental Genetics, Department of Biology, New York University, New York, United States

**Abstract** During embryogenesis, chromatin accessibility profiles control lineage-specific gene expression by modulating transcription, thus impacting multipotent progenitor states and subsequent fate choices. Subsets of cardiac and pharyngeal/head muscles share a common origin in the cardiopharyngeal mesoderm, but the chromatin landscapes that govern multipotent progenitors competence and early fate choices remain largely elusive. Here, we leveraged the simplicity of the chordate model *Ciona* to profile chromatin accessibility through stereotyped transitions from naive *Mesp*+ mesoderm to distinct fate-restricted heart and pharyngeal muscle precursors. An FGF-Foxf pathway acts in multipotent progenitors to establish cardiopharyngeal-specific patterns of accessibility, which govern later heart vs. pharyngeal muscle-specific expression profiles, demonstrating extensive spatiotemporal decoupling between early cardiopharyngeal enhancer accessibility and late cell-type-specific activity. We found that multiple *cis*-regulatory elements, with distinct chromatin accessibility profiles and motif compositions, are required to activate *Ebf* and *Tbx1/10*, two key determinants of cardiopharyngeal fate choices. We propose that these 'combined enhancers' foster spatially and temporally accurate fate choices, by increasing the repertoire of regulatory inputs that control gene expression, through either accessibility and/or activity.

**\*For correspondence:**
cr1636@nyu.edu (CR);
lc121@nyu.edu (LC)

**Competing interests:** The authors declare that no competing interests exist.

## Introduction

How a species' genome encodes its diverse and specific biological features has fascinated generations of biologists, and answers regarding the genetic control of body plan, organ, tissue and cell type formation have emerged from steady progress in developmental biology. Cell types arise as cells divide and the progeny of pluripotent embryonic stem cells progress through multipotent and fate-restricted states. The ontogeny of diverse terminal cell identities involves differential expression of hundreds to thousands of genes. Their dynamic activities are orchestrated by complex gene regulatory networks, whereby DNA-binding proteins and co-factors act upon specific *cis*-regulatory elements to control gene expression (*Davidson, 2010*). Technical and conceptual revolutions in genome biology have extensively characterized the chromatin dynamics that govern the function of *cis*-regulatory elements (*Klemm et al., 2019*). Specifically, as the nuclear genome is packaged in nucleosomes, DNA-binding transcription factors compete with histones to interact with *cis*-regulatory elements and control gene expression. Thus, identifying changing landscapes of accessible chromatin governing the transition from multipotent to fate restricted progenitors offers privileged insights into the genomic code for progressive cell type specification.

Dynamic chromatin states underlying cardiomyocyte differentiation have been extensively profiled (*Paige et al., 2012*; *Wamstad et al., 2012*), and chromatin state regulation is essential for heart development (*He et al., 2014*; *Rosa-Garrido et al., 2013*; *Zaidi et al., 2013*). However, different parts of the heart originate from separate first and second fields of progenitor cells, including those referred to as cardiopharyngeal, which can also produce branchiomeric head muscles (*Diogo et al., 2015*; *Lescroart et al., 2010*). Bulk and single cell transcription profiling have begun to illuminate

gene expression changes underlying cardiopharyngeal fate choices (*Lescroart et al., 2018*; *Lescroart et al., 2014*), but the corresponding chromatin dynamics remains largely elusive.

The tunicate *Ciona* emerged as a powerful chordate model to study early cardiopharyngeal development with high spatio-temporal resolution (*Diogo et al., 2015*; *Kaplan et al., 2015*). In *Ciona*, the cardiopharyngeal lineages arise from naive *Mesp+* mesodermal progenitors that emerge at the onset of gastrulation, and divide into two multipotent cardiopharyngeal progenitors (aka trunk ventral cells, TVCs) and two anterior tail muscles (ATMs), on either side of the embryo (*Figure 1A*). Following induction by FGF-MAPK signaling, cardiopharyngeal progenitors migrate collectively, before dividing asymmetrically and medio-laterally to produce small median first heart precursors (FHPs), and large lateral second trunk ventral cells (STVCs) (*Davidson et al., 2005*; *Stolfi et al., 2010*; *Wang et al., 2013*). The latter are also multipotent cardiopharyngeal progenitors, which upregulate *Tbx1/10* and then divide again to produce small median second heart precursors (SHPs), and

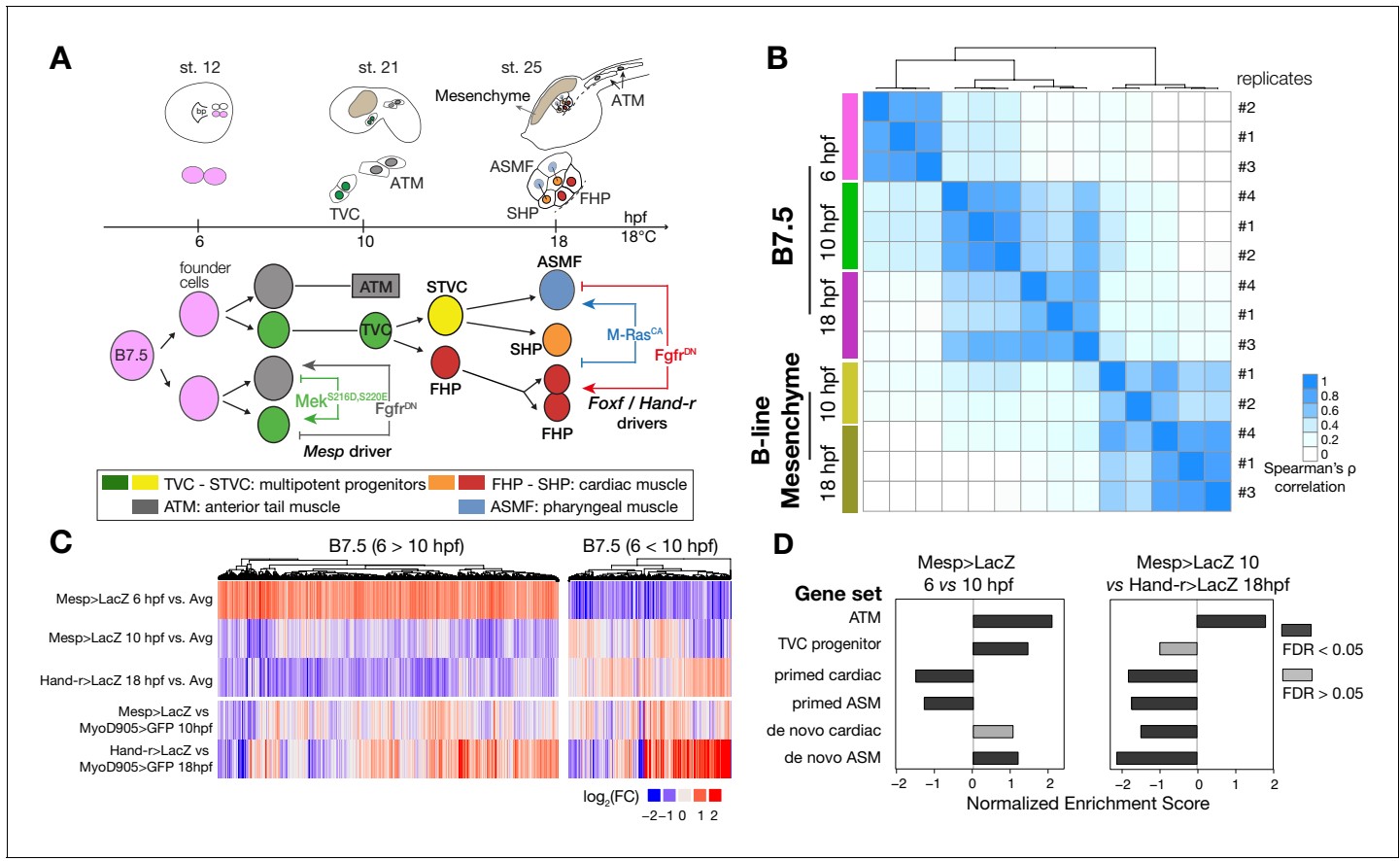

**Figure 1.** Profiling chromatin accessibility dynamics during early cardiopharyngeal cell development. (**A**) Embryos, larvae and lineage diagram showing B7.5 blastomeres, their cardiopharyngeal progeny, and the main stages sampled for ATAC-seq. Anterior tail muscle (ATM, gray), trunk ventral cell (TVC, green), secondary TVC (STVC, yellow), first heart precursor (FHP, red), second heart precursor (SHP, orange), atrial siphon precursor cells (ASMF, blue). Stages (St.) according to *Hotta et al. (2007)* with hours post fertilization (hpf). (**B**) Spearman correlation of RPKM (reads per kb per million mapped reads) values in 14,178 regions changing accessibility over time or between B7.5 and B-line mesenchyme lineages. (**C**) Temporal changes in chromatin accessibility for 5,450 regions. 'B7.5 6 > 10': 3,691 regions more accessible at *Mesp>LacZ* 6 hpf than *Mesp>LacZ* 10 hpf. 'B7.5 6 < 10': 1,759 regions more accessible at *Mesp>LacZ* 10 than *Mesp>LacZ* 6 hpf. The accessibility of these regions is shown for *Mesp>LacZ* 6 hpf, *Mesp>LacZ* 10 hpf, and *Hand-r>LacZ* 18 hpf vs. the average (avg) accessibility in the control cells. Cell-type-specific chromatin accessibility is shown in the comparison of *Mesp>LacZ* and *MyoD905>GFP* at 10 and *Hand-r>LacZ* and *MyoD905>GFP* 18 hpf. (**D**) Gene Set Enrichment Analysis (GSEA) normalized enrichment score of defined gene sets in regions ranked by difference in accessibility between time points as indicated (see Materials and methods).
The online version of this article includes the following figure supplement(s) for figure 1:

**Figure supplement 1.** General characterization of the accessome.

**Figure supplement 2.** Characterization of promoter regions.

**Figure supplement 3.** Annotation of the accessome.

large lateral atrial siphon muscle founder cells (ASMFs). ASMFs activate *Ebf,* which is necessary and sufficient to induce pharyngeal muscle specification (*Razy-Krajka et al., 2014*; *Stolfi et al., 2014*; *Stolfi et al., 2010*; *Tolkin and Christiaen, 2016*). Importantly, spatially and temporally accurate activation of *Tbx1/10* and *Ebf* in the STVC and ASMF, respectively, is essential to permit the emergence of all cardiopharyngeal cell lineages, as their ectopic expression would inhibit proper heart fate specification (*Figure 1A*).

Building on previous extensive transcription profiles (*Christiaen et al., 2008*; *Razy-Krajka et al., 2014*), including single cell RNA-seq (scRNA-seq) from multipotent cardiopharyngeal progenitors to first and second heart lineages and pharyngeal muscle precursors (*Wang et al., 2019*), here we characterized the genome-wide chromatin accessibility dynamics underlying cardiopharyngeal fate specification. We identified regulatory inputs that govern *cis*-regulatory element accessibility and activity, as well as cell-type-specific enhancers for key cardiopharyngeal determinants. We found that, in multipotent progenitors, an FGF-Foxf pathway controls cardiopharyngeal-specific patterns of accessibility, which govern later heart vs. pharyngeal muscle-specific expression profiles. We further characterized temporal patterns of chromatin accessibility during cardiopharyngeal development. In particular, activation of fate determinants *Tbx1/10* and *Ebf* specifically in the STVCs and ASMF, respectively, require multiple *cis*-regulatory elements with distinct spatio-temporal patterns of accessibility, which precede gene expression. We propose that these elements function as 'combined enhancers', which mediate distinct inputs, including from determinants of chromatin accessibility, to regulate gene activation. The observation that *cis*-regulatory inputs from multiple elements control expression of a single gene is consistent with the 'shadow-' and 'super-enhancer' paradigms (*Barolo, 2012*; *Hnisz et al., 2013*; *Hong et al., 2008*; *Kvon et al., 2014*; *Perry et al., 2011*; *Perry et al., 2010*; *Pott and Lieb, 2015*; *Whyte et al., 2013*). However, while shadow enhancer promotes robust transcription through the actions of multiple elements mediating similar regulatory inputs (*Frankel, 2012*; *Frankel et al., 2010*; *Lam et al., 2015*; *Zeitlinger et al., 2007*), we propose that combined enhancers promote spatially and temporally accurate fate choices, by augmenting the repertoire of *trans*-acting inputs controlling gene activation through enhancer activity and/or chromatin accessibility.

## Results

### A reference accessome for cardiopharyngeal development

To characterize the chromatin landscape underlying early cardiopharyngeal development, we used the assay for transposon-accessible chromatin (ATAC-seq; *Buenrostro et al., 2013*) on lineage-specific samples isolated at successive time points, and following defined perturbations (*Figure 1A*; *Supplementary file 1*; *Razy-Krajka et al., 2018*; *Wang et al., 2019*). Using the B7.5 lineage-specific *Mesp>tagRFP* reporter (*Wang et al., 2018*), we used FACS to collect ~4,000 cells per biological replicate from embryos dissociated at five time points encompassing key transitions in cardiopharyngeal development (*Figure 1A*): naive *Mesp+* mesoderm (aka founder cells; *Cooley et al., 2011*), ATMs, TVCs, STVCs as well as fate-restricted first and second heart precursors (FHPs and SHPs), and pharyngeal muscle precursors (aka atrial siphon muscle founder cells -ASMF- and their progeny, the ASM precursors -ASMP; *Razy-Krajka et al., 2014*). The latter fate-restricted progenitors were obtained from larvae dissociated at three time points (15, 18 and 20 hours post-fertilization, hpf; *Figure 1A*). From the same embryonic cell populations, we used a co-transfected *MyoD905>GFP* reporter to isolate B-line mesenchymal cells (*Christiaen et al., 2008*). In the present study, we predominantly focused our analysis on the cardiopharyngeal progenitors at 6, 10 and 18 hpf (*Figure 1A*).

We obtained ~500 million unique ATAC-seq reads, with fragment-size distributions showing the characteristic ~150 bp periodicity and patterns of mono-, di- and tri-nucleosomal fragments (*Buenrostro et al., 2013*), which were absent in the genomic DNA control (*Figure 1—figure supplement 1A*). We identified ATAC-seq peaks using MACS2 (*Zhang et al., 2008*), and generated a combined atlas of 56,090 unique and non-overlapping accessible regions covering 9.25% of the *C. robusta* genome, which we used as our reference 'accessome' (*Figure 1—figure supplement 1B,E*). General metrics including peak numbers, size, GC content and genomic distribution were comparable to consensus peaksets reported in other studies of chromatin accessibility in developmental

contexts (Materials and methods; *Figure 1—figure supplement 1*; *Daugherty et al., 2017*; *Hockman et al., 2019*; *Jänes et al., 2018*; *Li et al., 2007*; *Madgwick et al., 2019*).

Next, we annotated the reference accessome by associating accessible regions with other genomic features, especially gene models. In *Ciona*, the transcripts of approximately half of the protein-coding genes undergo spliced-leader (SL) *trans*-splicing, which replaces the original 5' end sequence of pre-mRNAs by a short non-coding RNA, causing the 5' end of mRNAs to differ from the transcription start site (TSS) (*Ganot et al., 2004*; *Hastings, 2005*; *Satou et al., 2006*; *Vandenberghe et al., 2001*). Using annotated TSSs (*Satou et al., 2006*; *Vandenberghe et al., 2001*; *Yokomori et al., 2016*), RNA-seq datasets (*Wang et al., 2019*), and our ATAC-seq data (*Figure 1—figure supplement 2C*), we determined that promoter regions and 5' untranslated regions (5'UTR) were over-represented in the accessome ($p < 0.001$, two-tailed binomial test; *Figure 1—figure supplement 1C*), indicating that promoter proximal regions tend to be accessible as observed in other systems (*Mayran et al., 2018*). We also detected nucleosome footprints immediately upstream of TSSs, consistent with a tendency for constitutive accessibility (*Figure 1—figure supplement 2A,B*; *Mavrich et al., 2008a*; *Mavrich et al., 2008b*). By contrast, intronic and intergenic regions were significantly under-represented in our reference accessome, compared to the whole genome, although they were the most abundant elements (32.8% and 20.8%, respectively; *Figure 1—figure supplement 1B*). This suggests that most of these elements are accessible in specific contexts, as expected for tissue-specific *cis*-regulatory elements (*Long et al., 2016*).

We associated annotated genes with ATAC-seq peaks located within 10 kb of the TSS or transcription termination site (TTS) (*Figure 1—figure supplement 3A*) (*Brozovic et al., 2018*), thus assigning median values of 11 peaks per gene, and three genes per peak, owing to the compact *Ciona* genome (*Figure 1—figure supplement 3B,C*). Notably, active regulatory genes encoding transcription factors and signaling molecules were associated with significantly more peaks than other expressed genes ($p < 0.001$, two-tailed binomial test; *Figure 1—figure supplement 3F*). This high peak density surrounding regulatory genes is reminiscent of previously described super-enhancers (*Whyte et al., 2013*) and Clusters of Open *Cis*-Regulatory Elements (COREs) surrounding developmental regulators (*Gaulton et al., 2010*; *Khan et al., 2018*; *Pott and Lieb, 2015*).

## Cardiopharyngeal accessibility profiles are established in multipotent progenitors

Using this reference accessome, we investigated lineage-specific and dynamic patterns of chromatin accessibility during fate decisions. We observed the greatest contrast in accessibility between the B7.5 and B-line mesenchyme lineages, with biological replicates correlating most highly (Spearman's ρ > 0.93), indicating reproducible detection of extensive lineage-specific accessibility (*Figure 1B*). Within the B7.5 lineage, correlation analysis suggested that most changes occur between 6 and 10 hpf, during the transition from naive *Mesp+* mesoderm to multipotent cardiopharyngeal progenitors (TVCs). Higher correlation between multipotent progenitors and mixed heart and pharyngeal muscle precursors, obtained from 18 hpf larvae, suggested more stable accessibility profiles during and immediately following early cardiopharyngeal fate choices (*Figure 1B*). Consistent with correlation analyses, most significant temporal changes in accessibility occurred during the transition from naive *Mesp+* mesoderm to multipotent progenitors (5,450 regions, FDR < 0.05; *Figure 1C*). Specifically, about two thirds (64.7%, 3,525/5,450) of these regions showed reduced accessibility at 10 hpf, in multipotent progenitors, compared to 6 hpf naive *Mesp+* mesoderm (*Figure 1C*). Conversely, 1,252 regions become accessible between 6 and 10 hpf or later, and 38.8% (486/1,252) of these regions were more accessible in the B7.5-lineage compared to the mesenchyme (*Figure 1C*). Moreover, the subset of regions opening between 6 and 10 hpf or later was enriched in genomic elements associated with cardiopharyngeal markers, including primed pan-cardiac and pharyngeal muscle markers, while elements flanking tail muscle markers (ATMs) or multipotent progenitor-specific genes were predominantly closing between 6 and 10 to 18 hpf (*Figure 1D*; see Materials and methods for definition of gene sets). Taken together, these observations suggest that cardiopharyngeal accessibility profiles are established specifically in the B7.5 lineage, upon induction of multipotent progenitors, and persist in fate-restricted cells.

To further analyze changes in accessibility associated with multipotent progenitor induction, we performed ATAC-seq on B7.5 lineage cells isolated at 10 hpf following defined perturbations of FGF-MAPK signaling: a constitutively active form of Mek ($Mek^{S216D,S220E}$), which converts all B7.5

lineage cells into multipotent cardiopharyngeal progenitors or a dominant negative form of the Fgf receptor (*Fgfr^DN*), which blocks induction and transforms all B7.5-derived cells into ATMs (*Figure 1A*; *Davidson et al., 2006*; *Razy-Krajka et al., 2018*). We used DESeq2 (*Love et al., 2014*) to compute differential accessibility of the elements in the reference accessome, and identified 2,728 and 2,491 differentially accessible regions following either inhibition or activation of FGF-MAPK signaling, respectively (*Figure 2A,B*; *Figure 2—figure supplement 1A*). Using peak-to-gene annotations (*Figure 1—figure supplement 3A*), we cross-referenced ATAC-seq with expression microarray data obtained from B7.5 lineage cells expressing the same *Fgfr^DN*(*Christiaen et al., 2008*), and observed a positive correlation between changes in differential accessibility and differential gene expression at 10 hpf (Spearman's ρ = 0.47; *Figure 2A*; *Figure 2—figure supplement 2A*). Specifically, 48% of FGF-MAPK-regulated genes were associated with at least one element showing consistent differential accessibility, including 260 candidate FGF-MAPK-activated TVC markers associated with 557 regions predicted to open specifically in multipotent cardiopharyngeal progenitors at 10 hpf (*Supplementary file 2*, *Figure 2—figure supplement 2B*). Conversely, the majority (603 regions) of differentially accessible ATAC-seq peaks associated with 263 FGF-MAPK-inhibited tail muscles markers, and were also more accessible upon inhibition of FGF signaling (*Supplementary file 2*; *Figure 2—figure supplement 1A*). Taken together, these observations indicate that cardiopharyngeal accessibility profiles are established in the multipotent progenitors by opening regions associated with genes upregulated upon induction by FGF-MAPK signaling.

Consistent with the hypothesis that FGF-MAPK-dependent, cardiopharyngeal-specific elements act as tissue-specific enhancers, they were predominantly found in intronic or intergenic regions (48% and 37%, respectively, FDR < 0.05, one-tailed hypergeometric test). Conversely, tissue-specific peaks associated with tail muscle markers were enriched in promoters, TSS and 5'UTR (one-tailed hypergeometric test, FDR < 0.05, 57%, 23% and 15%, respectively; *Figure 2—figure supplement 2C*). Previously characterized enhancers for TVC-specific genes *Lgr4/5/6*, *Rhodf*, *Foxf*, *Unc5*, *Rgs21*, *Ddr*, *Asb2* and *Gata4/5/6* showed ATAC-seq patterns consistent with cardiopharyngeal-specific accessibility (*Supplementary file 3*; *Beh et al., 2007*; *Bernadskaya et al., 2019*; *Christiaen et al., 2008*; *Woznica et al., 2012*). We thus leveraged differential accessibility profiles to identify novel enhancers of cardiopharyngeal gene expression. We focused on a locus containing the conserved cardiac determinant and TVC marker, *Nk4/Nkx2-5* (*Wang et al., 2013*), and two tail-muscle specific *Myosin regulatory light chain* (*Mrlc*) genes (*Kusakabe et al., 2004*; *Satou et al., 2001b*; *Sierro et al., 2006*), with associated elements showing the predicted TVC- and ATM-specific accessibility patterns, respectively (*Figure 2A,D*). Reporter gene expression assays showed that a DNA fragment containing differentially accessible elements located in the *Nk4/Nkx2-5* intron (*KhC8.2200* and *.2201*) was sufficient to drive *GFP* expression specifically in cardiopharyngeal multipotent progenitors (*Figure 2E*). B7.5 lineage-specific CRISPR/Cas9-induced deletions of these elements reduced or eliminated *Nk4/Nkx2-5* expression specifically in TVCs, thus demonstrating its role as a *bona fide* cardiopharyngeal enhancer (*Figure 2F*; *Figure 2—figure supplement 3D*). Extending these analyses to other loci, including *Fgf4*, *Fzd4*, *Foxg-r*, *Fbln*, *Eph1*, *Ncaph*, *Hand* and *Smurf1/2*, we identified 8 out of 15 candidate cardiopharyngeal enhancers that drove reporter expression in the multipotent progenitors (*Figure 2—figure supplement 4*; *Supplementary file 4*), and B7.5-lineage-specific CRISPR/Cas9-mediated mutagenesis targeting differentially accessible elements reduced TVC-specific expression of the neighbouring genes *Fgf4*, *Smurf1/2* and *Fbln* (*Figure 2—figure supplement 5*; *Figure 2—figure supplement 6*; *Supplementary file 5*). Conversely, candidate ATM-specific elements activated reporter gene expression in the tail muscles, including ATM cells, but not in the cardiopharyngeal progenitors, and were located near tail muscle markers (*Figure 2—figure supplement 1B,C*; *Supplementary file 6*). Collectively, these findings indicate that genomic elements that open specifically in multipotent progenitors act as transcriptional enhancers of cardiopharyngeal gene expression and their accessibility is controlled by FGF-MAPK induction.

## A Foxf-dependent code for cardiopharyngeal accessibility

Next, we harnessed chromatin accessibility patterns predictive of cardiopharyngeal enhancer activity to identify enriched sequence motifs, and thus candidate regulators of chromatin accessibility and gene expression. For this, we performed a one-tailed hypergeometric test for enrichment of known motifs. We complemented this analysis by calculating differential accessibility of motifs using chromVAR (*Schep et al., 2017*), which was developed to analyze sequence motifs associated with cell-

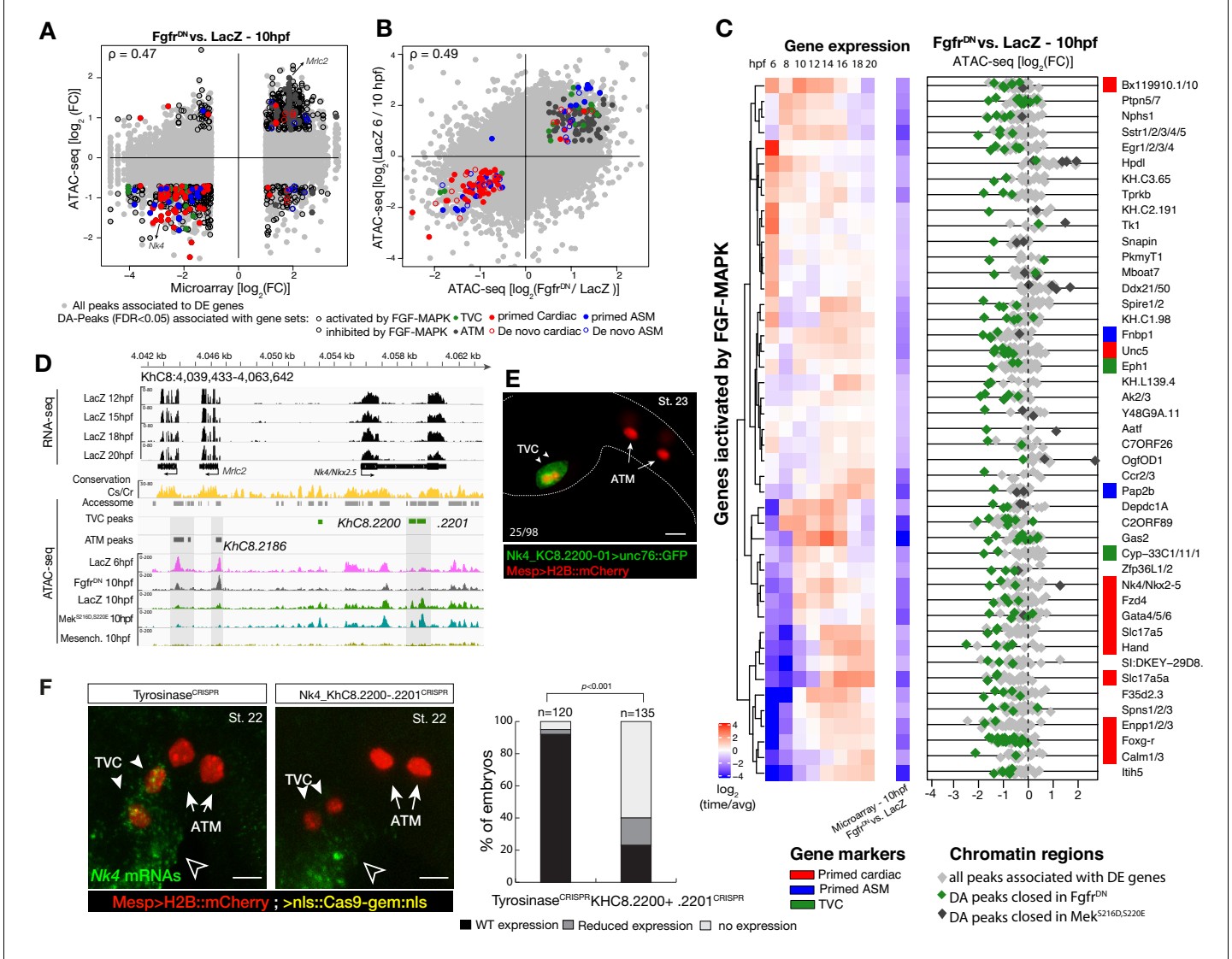

**Figure 2.** Cardiopharyngeal accessibility profiles are established in multipotent progenitors. (A–B) Correlations between differential gene expression (DE) and differential chromatin accessibility (DA) in response to FGF-MAPK perturbation in the multipotent progenitors (A) and between chromatin accessibility in response to FGF-MAPK perturbation and in multipotent progenitors (10 hpf) versus founder cells (6 hpf). (B). Colored dots are DA peaks associated with cell type-specific DE genes. ρ is the Spearman correlation of expression and accessibility for DA regions associated with DE genes (A) or of region response to MAPK perturbation with accessibility in founder cells versus multipotent progenitors (B). (C) Relationship between expression and accessibility of DE genes associated with DA regions for genes in the bottom 0.75% quantile of fold change between expression in $Fgfr^{DN}$ and control ($\log_2$(FC) < −1.32). Microarray $\log_2$(fold change (FC)) values are shown on the left. The fold change for all time points is versus the average. (D) A 24 kb region on chromosome eight displaying expression (RNA-seq) and chromatin accessibility (ATAC-seq; normalized by total sequencing depth). Gray shaded boxes show validated ATM-specific promoters and a newly identified TVC-specific enhancer in $Nk4/Nkx2-5$ intron. (E) Enhancer-driven in vivo reporter expression (green) of tested ATAC-seq regions (KhC8.2200 and .2201). TVCs marked with $Mesp>H2B::mCherry$ (red). Numbers indicate observed/total of half-embryos scored. (F) Endogenous expression of $Nk4/Nkx2-5$ visualized by in situ (green) in $Tyrosinase^{CRISPR}$ and upon CRISPR/Cas9-induced deletions of TVC-specific region. Nuclei of B7.5 lineage cells are labelled by $Mesp>nls::LacZ$ and revealed with an anti beta-galactosidase antibody (red). $Nk4/Nkx2-5$ expression was not affected in the epidermis (open arrowhead). Experiment performed in biological replicates. Scale bar, 20 μm. Fisher exact test, total numbers of individual halves scored per condition are shown in 'n='. Gene expression data for 6 hpf and 'FGF-MAPK perturbation 10 hpf' (*Christiaen et al., 2008*) and 8 to 20 hpf (*Razy-Krajka et al., 2014*) were previously published.

The online version of this article includes the following figure supplement(s) for figure 2:

**Figure supplement 1.** Inhibition of FGF signaling ($Mesp>Fgfr^{DN}$−10 hpf) induces opening of ATM-specific elements.
**Figure supplement 2.** General characterization of differential accessibility.
**Figure supplement 3.** Peakshift validation of sgRNA efficiency.
**Figure supplement 4.** Candidate TVC-specific enhancers in vivo validation by reporter gene assay.

*Figure 2 continued on next page*

*Figure 2 continued*

**Figure supplement 5.** Candidate TVC-specific enhancers in vivo validation by CRISPR/Cas9.
**Figure supplement 6.** CRISPR validation on the TVC-specific Fgf4 enhancer.

type-specific accessibility (*Figure 3A*). Naive *Mesp+* mesoderm-specific elements, which closed between 6 and 10 hpf, were enriched in motifs for Homeodomain, T-box and Ets families of transcription factors (TF), consistent with documented roles for Lhx3/4, Tbx6 and Ets homologs in B7.5 blastomeres (*Figure 3A*; *Davidson et al., 2006*; *Davidson et al., 2005*; *Satou et al., 2001a*). Candidate tail muscle-specific elements, which opened upon *Fgfr^DN* misexpression, were similar to naive *Mesp+* mesoderm, and enriched in motifs for the basic helix-loop-helix (bHLH) family of TFs, which includes Mesp and Mrf/MyoD, a conserved muscle-specific transcription regulator that promotes tail muscle differentiation (*Christiaen et al., 2008*; *Meedel et al., 2007*; *Razy-Krajka et al., 2014*; *Tolkin and Christiaen, 2016*). By contrast, motifs for Zinc Finger, Fox/Forkhead and nuclear receptor families of TFs were enriched among candidate cardiopharyngeal-specific elements, revealing a typical mesendodermal signature for early cardiopharyngeal progenitors (*Cusanovich et al., 2018*).

Combined with motif enrichment analyses, temporal gene expression profiles (*Razy-Krajka et al., 2014*) identified candidate *trans*-acting regulators of cardiopharyngeal-specific accessibility and/or activity (*Figure 3B*). For example, regions specifically accessible in tail muscle- or naive *Mesp+* mesoderm were enriched in homeobox, Ets and T-box motifs, consistent with early expression of *Mrf/MyoD*, *Lhx3/4*, *Ets1/2*, and *Tbx6*, respectively. Similarly, the increased accessibility of motifs for K50 Paired homeodomain proteins in naive *Mesp+* mesoderm indicated a possible role for *Otx*, which is expressed early in B7.5 blastomeres (*Figure 3A,B*; *Hudson et al., 2003*). Cardiopharyngeal-specific enrichment for Fox/Forkhead and Zinc Finger motifs pointed to several known factors, including Foxf, one of the first genes activated in multipotent progenitors upon induction by FGF-MAPK, prior to *Gata4/5/6* (*Beh et al., 2007*; *Christiaen et al., 2008*; *Ragkousi et al., 2011*). Moreover, GATA and Forkhead proteins are founding members of a group of TFs known as pioneers, which can bind their target sites in closed chromatin and promote accessibility (*Cirillo et al., 2002*; *Zaret and Carroll, 2011*). Protein sequence alignments indicated the presence of key residues, conserved between the DNA binding domains of Foxf and the classic pioneer FOXA, which mimic linker histone H1 in its ability to displace DNA-bound nucleosomes (*Figure 3—figure supplement 1B,C*; *Clark et al., 1993*). Finally, the *Foxf* enhancer was accessible in naive *Mesp+* founder cells, suggesting that it is poised for activation, unlike the intronic *Gata4/5/6* enhancer (*Figure 3—figure supplement 2A*). Consistent with a role for Fox and GATA proteins in opening and activating the *Nk4/Nkx2-5* enhancer, we found putative cognate binding sites in the newly identified element to be conserved with the closely related species, *C. savignyi* (*Figure 3—figure supplement 3B,C*). Taken together, these analyses identified a putative code for cardiopharyngeal-specific accessibility and enhancer activity, which comprise motifs for candidate DNA binding factors of the Forkhead and GATA and identified *Foxf* as candidate determinant of cardiopharyngeal accessibility.

To test if Foxf contributes to establishing cardiopharyngeal accessibility and gene expression profiles, we used reagents for B7.5 lineage-specific loss-of-function by CRISPR/Cas9-mediated mutagenesis (*Foxf^CRISPR*; *Gandhi et al., 2017*), and performed ATAC- and RNA-seq on FACS-purified cells isolated from tailbud embryos at 10 hpf. RNA-seq confirmed that CRISPR/Cas9 mutagenesis inhibited *Foxf* itself and other TVC-expressed genes, including effectors of collective cell migration such as *Ddr*, consistent with previous microarray data (*Bernadskaya et al., 2019*; *Christiaen et al., 2008*) (*Figure 3D*; *Figure 3—figure supplement 2B*). Out of 52 differentially expressed genes (*Figure 3—figure supplement 2B*; *Supplementary file 7*), seven down-regulated genes were previously annotated as primed pan-cardiac markers, including *Hand*, *Gata4/5/6* and *Fzd4* (*Wang et al., 2019*). Down-regulated genes also included primed pharyngeal muscle markers, such as *Rhod/f* (*Figure 3D*; *Figure 3—figure supplement 2B*; *Christiaen et al., 2008*; *Razy-Krajka et al., 2014*), suggesting that Foxf promotes the onset of both the cardiac and pharyngeal muscle programs in multipotent progenitors, a feature known as multilineage transcriptional priming (*Razy-Krajka et al., 2014*; *Wang et al., 2019*).

Consistent with the effects of *Foxf* mutagenesis on gene expression, regions closed in *Foxf^CRISPR* samples included known cardiopharyngeal enhancers for *Gata4/5/6* and *Ddr* (*Figure 3D*; *Figure 3—*

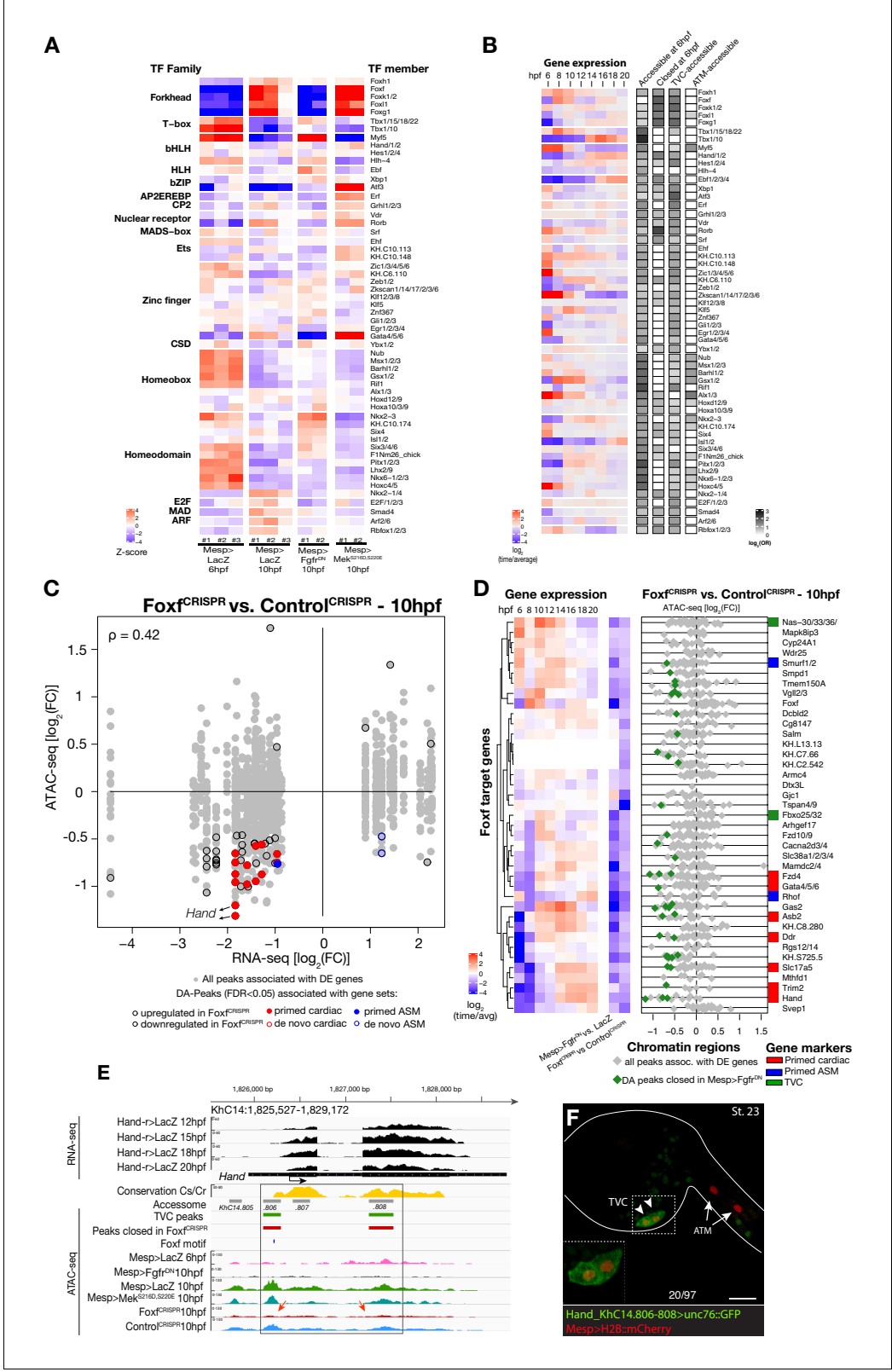

**Figure 3.** Foxf is required for cardiopharyngeal-specific chromatin accessibility. (**A**) Motif accessibility between libraries from chromVAR (*Schep et al., 2017*). Motifs were obtained and associated with *Ciona* transcription factors (TFs) as described in the Materials and methods. Deviations were computed for FGF signaling-dependent regions at 10 hpf and B7.5 replicates at 6 and 10 hpf. We calculated the differential accessibility of all motifs between conditions and time points. Only the most significant motif is shown for each TF. (**B**) Expression of transcription factors over time compared to

*Figure 3 continued*

enrichment of corresponding TF motifs in condition specific peak sets. log$_2$(odds ratio) values (log$_2$(OR), see Materials and methods) are shown for motifs that are significantly enriched in a peak set (one-tailed hypergeometric test, FDR < 0.05). Only TFs expressed in the B7.5 lineage are shown. (C) Differential expression of Foxf target genes (DE) vs. differential chromatin accessibility (DA) in *Foxf*$^{CRISPR}$. ρ is the Spearman correlation of expression and accessibility for DA regions associated with DE genes. (D) Association between expression of Foxf target genes and accessibility of proximal regions which were both TVC-specific and closed in *Foxf*$^{CRISPR}$ as in *Figure 2C*. (E) A 3.6 kb region on chromosome 14 displaying expression profiles of RNA-seq and chromatin accessibility profiles of ATAC-seq normalized tag count. Foxf core binding site (GTAAACA) is displayed as blue line. The boxed region indicates a newly identified TVC-specific enhancer in *Hand* locus. Red arrow indicates a TVC-specific enhancer showing closed chromatin in *Foxf*$^{CRISPR}$ ATAC-seq. (F) Enhancer-driven in vivo reporter expression (green) of tested ATAC-seq peaks. TVCs marked with *Mesp>H2B::mCherry* (red). Numbers indicate observed/total of half-embryos scored. Experiment performed in biological replicates. Scale bar, 30 μm. Gene expression data for 6 hpf and 'FGF-MAPK perturbation 10 hpf' (*Christiaen et al., 2008*), and from 8 to 20 hpf (*Razy-Krajka et al., 2014*) were previously published. The online version of this article includes the following figure supplement(s) for figure 3:

**Figure supplement 1.** Conservation of DNA-recognition motifs of Foxf proteins.
**Figure supplement 2.** General characterization of *Foxf*$^{CRISPR.}$
**Figure supplement 3.** Conserved binding motifs in TVC-specific *Nk4/Nkx2-5* enhancer.
**Figure supplement 4.** *Foxf* loss-of-function (*Foxf*$^{CRISPR}$) caused closing of TVC-specific enhancers.
**Figure supplement 5.** Conserved binding motifs in TVC-specific *Hand* enhancer.
**Figure supplement 6.** Conserved binding motifs in *Smurf* enhancer.
**Figure supplement 7.** Accessible elements annotated to Foxf target genes.

*figure supplement 4*; *Bernadskaya et al., 2019*; *Christiaen et al., 2008*; *Woznica et al., 2012*), newly identified enhancers for *Eph1*, *Smurf1/2* and *Fzd4*, and a novel enhancer of *Hand* expression (*Hand_KhC14.805 -. 807*; *Figure 3C–F*, *Supplementary file 8*). These differentially accessible elements contain several, evolutionary conserved, putative Fox binding sites (*Figure 3—figure supplement 5*; *Figure 3—figure supplement 6*). We identified two conserved putative Forkhead binding sites in the minimal STVC-specific enhancer from the *Tbx1/10* locus (termed *T12*; *Razy-Krajka et al., 2018*), which were necessary for reporter gene expression (*Figure 5—figure supplement 1D,E*). Moreover, loss-of-function of *Foxf* (*Foxf*$^{CRISPR}$) drastically reduced *T12* enhancer activity (*Figure 5—figure supplement 1*). These results are consistent with the hypothesis that Foxf acts directly on the minimal *Tbx1/10* enhancer to promote its activity in the second multipotent cardiopharyngeal progenitors.

Notably, 98% (40/41) of the regions with diminished accessibility following *Foxf* inhibition, and located near a candidate Foxf target gene, were also accessible in multipotent progenitor cells (*Figure 3D*; *Figure 3—figure supplement 7A*). Moreover, 22% (600/2,728) of the predicted multipotent progenitor-specific elements were closed upon *Foxf* inhibition, and gene set enrichment analysis indicated that *Foxf* loss-of-function generally decreased the accessibility of cardiopharyngeal-specific elements (*Figure 3—figure supplement 2C*; *Figure 3—figure supplement 7B*). Finally, 18 of 41 (44%) of the *Foxf*-dependent elements associated with candidate Foxf targets were closed in 6 hpf founder cells and appear to open specifically in the cardiopharyngeal progenitors by 10 hpf (*Figure 3—figure supplement 2D*; *Figure 3—figure supplement 7B,C*; *Supplementary file 9*). This dynamic is consistent with a requirement for Foxf activity following its activation in the TVCs, immediately after division of the naive *Mesp*+ progenitors. Taken together, these results indicate that, in newborn multipotent progenitors, FGF-MAPK signaling upregulates *Foxf* (*Beh et al., 2007*; *Christiaen et al., 2008*), which is in turn required to open a substantial fraction of cardiopharyngeal-specific elements for gene expression in multipotent progenitors, including for such essential determinants as *Gata4/5/6* and *Hand* (*Figure 4H*).

## Chromatin accessibility in late heart vs. pharyngeal muscle precursors

Besides controlling coherent chromatin opening, enhancer activity and gene expression in multipotent cardiopharyngeal progenitors, FGF-Foxf inputs also appeared to open regions associated with later de novo-expressed heart and pharyngeal muscle markers (*Figure 2A,B*; *Supplementary file 10*; *Razy-Krajka et al., 2018*; *Razy-Krajka et al., 2014*; *Wang et al., 2019*; *Wang et al., 2013*). Accessibility patterns were also better correlated between 10 and 18 hpf (*Figure 1B*), suggesting a decoupling between early accessibility and late heart- vs. pharyngeal muscle-specific expression in late fate-restricted precursors. To identify accessibility patterns underlying the heart vs. pharyngeal

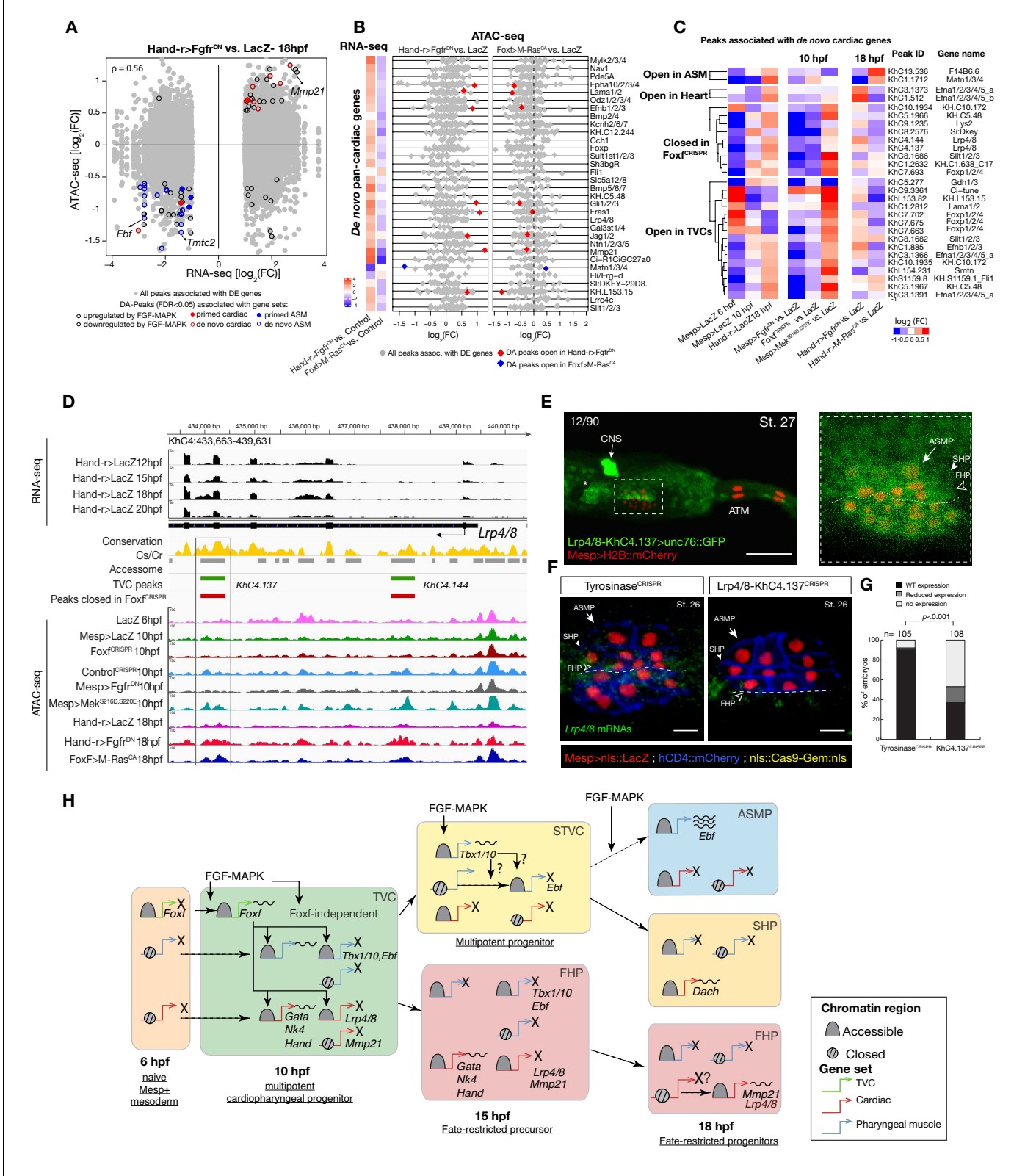

**Figure 4.** Cardiopharyngeal lineage-specific accessibility profiles and decoupling between enhancer accessibility and activity for de novo expressed genes. (**A**) Differentially expressed (DE) genes vs. differentially accessible (DA) peaks in response to FGF-MAPK perturbation in the fate-restricted cells. ρ is the Spearman correlation of expression and accessibility for DA peaks associated with DE genes. (**B**) Relationship between accessibility and expression of de novo pan-cardiac genes as in **Figure 2C**. DE genes in either condition are shown on the left. (**C**) Time-dependent ATAC-seq peaks

*Figure 4 continued on next page*

Figure 4 continued

associated with de novo expressed pan-cardiac genes. The accessibility of these peaks is shown for 6, 10 and 18 hpf vs. the average accessibility in the controls (*LacZ*) and upon FGF-MAPK perturbations at either 10 or 18 hpf. Peaks were classified as 'Open in ASM' (less accessible in *Fgfr*$^{DN}$ vs. *M-Ras*$^{CA}$ or *LacZ* at 18 hpf), 'Open in Heart' (less accessible in *M-Ras*$^{CA}$ vs. *Fgfr*$^{DN}$ or *LacZ* at 18 hpf), 'Closed in *Foxf*$^{CRISPR}$' (less accessible in *Foxf*$^{CRISPR}$ vs. *Control*$^{CRISPR}$), or 'Open in TVC' (less accessible in *Fgfr*$^{DN}$ vs. *Mek*$^{S216D,S220E}$ or *LacZ* at 10 hpf). Only regions changing accessibility between 6 and 10 hpf, or 10 and 18 hpf are shown. (D) A 6 kb region on chromosome four displaying expression profiles of RNA-seq and chromatin accessibility profiles of ATAC-seq normalized tag count. Peak ID refers to elements tested for reporter assay in vivo. The newly identified enhancer in *Lrp4/8* locus is in the boxed region. (E) Enhancer-driven in vivo reporter expression (green) of tested '*KhC4.137*' peak. TVCs marked with *Mesp>H2B::mCherry* (red). Numbers indicate observed/total of half-embryo scored. Zoom on cardiopharyngeal cell lineage (panel on the right). (F) Endogenous expression of *Lrp4/8* visualized by in situ (green) in *Tyrosinase*$^{CRISPR}$ and upon CRISPR/Cas9-induced deletion of ATAC-seq peaks. Nuclei of B7.5 lineage cells are labelled by *Mesp>nls::LacZ* and revealed with an anti beta-galactosidase antibody (red). *Mesp*-driven hCD4::mCherry accumulates at the cell membrane as revealed by anti mCherry antibody (Blue). Experiment performed in biological replicates. Scale bar = 10 μm. (G) Fisher exact test; n is the total number of individual embryo halves scored per condition. (H) Summary model: patterns of chromatin accessibility dynamics and gene expression during early cardiopharyngeal fate specification.

The online version of this article includes the following figure supplement(s) for figure 4:

**Figure supplement 1.** General characterization of FGF-MAPK perturbation.
**Figure supplement 2.** Differential accessibility in response to FGF/MAPK perturbation at 18 hpf.
**Figure supplement 3.** Accessibility of elements annotated to de novo ASM genes.
**Figure supplement 4.** Accessibility of binding motifs over time for elements annotated to de novo-expressed genes.

muscle fate choices, we compared bulk RNA-seq (*Wang et al., 2019*) and ATAC-seq datasets obtained from cardiopharyngeal lineage cells isolated from 18 hpf larvae, following the same defined perturbations of FGF-MAPK signaling (*Figure 1A*; *Figure 4A*; *Figure 4—figure supplement 1A–D*; *Figure 4—figure supplement 2A,B*; *Davidson et al., 2006*; *Razy-Krajka et al., 2018*). Among cardiac and pharyngeal muscle markers, we identified 35 FGF-MAPK-regulated genes associated with one or more elements showing consistent differential accessibility (*Figure 4A,B*; *Figure 4—figure supplement 2A,B*; *Supplementary file 11*). This indicated that, at least for a subset of cardiopharyngeal marker genes, FGF-MAPK-dependent changes in gene expression follow corresponding changes in chromatin accessibility in early heart and pharyngeal muscle precursors.

Gene-level inspection of differential accessibility associated with either inhibition or activation of gene expression revealed that only a fraction of associated elements was either closing or opening upon perturbation of FGF-MAPK signaling (*Figure 4B*; *Figure 4—figure supplement 2B*). For example, the first heart lineage marker _Matrix metalloproteinase 21/Mmp21_ (*Wang et al., 2019*) was associated with multiple upstream and intronic elements, but only some of these elements were differentially accessible following either gain or loss of FGF-MAPK function (*Figure 4—figure supplement 2A–C*), and a 3 kb fragment containing the upstream differentially accessible element sufficed to drive reporter gene expression throughout the cardiopharyngeal lineage, but not specifically in the first heart precursors (*Figure 4—figure supplement 2C–E*). Similarly, reporter gene expression assays showed that DNA fragments containing differentially accessible elements located ~0.5 kb upstream of the coding region of the de novo-expressed gene *Tmtc2* (*KhC2.3468*), and upstream of *KH.C1.1093_ZAN* (*KhC3.47, KhC3.46*), were sufficient to drive *GFP* expression in both cardiac and pharyngeal muscle progenitors, consistent with the notion that electroporated plasmids are not 'chromatinized' and thus constitutively accessible (*Figure 4—figure supplement 2F–G*; *Figure 4—figure supplement 3C–D*). This suggested that, for genes like *Mmp21*, *Tmtc2* and *Zan*, cell-type-specific accessibility determines cardiac vs. pharyngeal muscle-specific gene expression.

Remarkably, the vast majority (91%, 356 genes out of 391, *Supplementary file 12*) of differentially expressed genes were not associated with differentially accessible elements (*Figure 4A,B*; *Figure 4—figure supplement 2A,B*). Specifically, out of 30 de novo-expressed pan-cardiac genes that were also differentially expressed upon FGF-MAPK perturbation at 18 hpf, only 8 (27% ± 8%, SE) were associated with one differentially accessible element following perturbation of FGF-MAPK signaling (*Figure 4B*). Similarly, out of 23 de novo-expressed pharyngeal muscle genes, which were also differentially expressed upon FGF-MAPK perturbation at 18 hpf, 11 were associated with one differentially accessible element following perturbation of FGF-MAPK signaling (*Figure 4—figure supplement 3A*). This suggested that most differential gene expression in early heart and pharyngeal muscle precursors arise from differential *cis*-regulatory activity of elements that are otherwise

accessible throughout the cardiopharyngeal mesoderm. In keeping with this hypothesis, accessible regions associated with de novo-expressed pan-cardiac and pharyngeal muscle markers tended to open between 6 and 10 hpf, in a pattern consistent with FGF- and Foxf-dependent cardiopharyngeal-specific accessibility (*Figure 4C*; *Figure 4—figure supplement 3B*). These observations suggest that *cis*-regulatory elements controlling cell-type-specific de novo gene expression open in multipotent progenitors, prior to becoming active in fate-restricted precursors. Such decoupling between enhancer accessibility and activity has been observed in other developmental contexts, including early cardiogenesis in mammals (*Paige et al., 2012*; *Wamstad et al., 2012*).

As a proof of principle, we analyzed the *Lrp4/8* locus, which harbors two intronic elements (*KhC4.137* and *KhC4.144*) that opened upon TVC induction in an FGF- and Foxf-dependent manner, prior to *Lrp4/8* upregulation in cardiac progenitors (*Wang et al., 2019*), and were not differentially accessible at 18 hpf (*Figure 4C,D*). Of the two regions, only *KhC4.137* was sufficient to drive *GFP* expression in heart precursors indicating enhancer activity, and illustrating the decoupling between early and broad accessibility and late, cell-type-specific, activity (*Figure 4E*). Reporter gene expression and CRISPR/Cas9-mediated mutagenesis assays followed by FISH indicated that *KhC4.137* is both necessary and sufficient to activate gene expression in heart precursors (*Figure 4E,F*), showing that it acts as a *bona fide* enhancer, and demonstrating a specific case of decoupling between early and broad accessibility and late, cell-type-specific, activity.

To identify candidate regulators of late accessibility and/or activity, we parsed accessible elements associated with de novo-expressed heart and pharyngeal muscle markers into pre-accessible/primed or de novo-accessible elements and discovered sequence motifs enriched in each category (*Figure 4—figure supplement 4A*; *Supplementary file 13*). Putative binding sites for SMAD and homeodomain proteins such as Smad4 and Pitx respectively were enriched among pre-accessible elements associated with cardiac markers, and found in the primed elements regulating *Lrp4/8* upregulation (*Figure 4—figure supplement 4A,B*), suggesting a specific role in transcriptional activation, consistent with conserved roles for Pitx2 and BMP-SMAD signaling during heart development (*Figure 4—figure supplement 4C,D*; *Nowotschin et al., 2006*; *Schultheiss et al., 1997*). Motifs for known regulators of cardiac development, including Meis (*Desjardins and Naya, 2016*; *Paige et al., 2012*), were over-represented among de novo-accessible elements associated with cardiac markers, suggesting roles in establishing accessibility and/or regulating enhancer activity (*Figure 4—figure supplement 4A*). Notably, GATA motifs were enriched in primed accessible elements associated with cardiac markers, consistent with conserved roles for GATA factors as pioneer factors, and during cardiac development (*Pikkarainen et al., 2004*). Among motifs enriched in accessible elements associated with de novo-expressed pharyngeal muscle markers, the presence of ETS-, bHLH, and EBF-family motifs is consistent with established roles for FGF-MAPK, Hand-r, Mrf and Ebf in pharyngeal muscle specification (*Razy-Krajka et al., 2018*; *Razy-Krajka et al., 2014*; *Stolfi et al., 2014*; *Stolfi et al., 2010*; *Wang et al., 2013*). Notably, the enrichment of Ebf motifs among de novo-accessible elements associated with de novo-expressed pharyngeal muscle markers is reminiscent of the ability of EBF-family factors to interact with nucleosome-bound cognate sites, suggestive of a pioneering activity in committed pharyngeal muscle precursors (*Boller et al., 2016*; *Buenrostro et al., 2013*). In summary, this analysis identified distinct combinations of established and putative *trans*-acting factors differentially controlling the accessibility and/or activity of *cis*-regulatory elements that govern heart- vs. pharyngeal-muscle-specific gene expression (*Figure 4H*).

## Combinatorial *cis*-regulatory control of cardiopharyngeal determinants

The above analyses focused on one-to-one associations between accessible elements and neighboring genes to uncover candidate *trans*-acting inputs controlling gene expression through defined elements. However, most genes are associated with multiple accessible regions, especially developmental regulators (*Figure 1—figure supplement 3D–F*), presumably exposing diverse motifs for transcription factor binding. Moreover, distinct elements associated with the same neighboring gene often exhibited different accessibility dynamics. For instance, the loci of several de novo-expressed heart and pharyngeal muscle markers contained both primed-accessible and de novo-accessible elements (*Figure 4C*; *Figure 4—figure supplement 3B*; *Figure 4—figure supplement 4A,B*). This suggested that individual genes respond to a variety of regulatory inputs mediated through separate *cis*-regulatory elements.

To explore this possibility, we focused on *Tbx1/10* and *Ebf*, two established determinants of cardiopharyngeal fates (*Razy-Krajka et al., 2018*; *Razy-Krajka et al., 2014*; *Stolfi et al., 2014*; *Stolfi et al., 2010*; *Tolkin and Christiaen, 2016*; *Wang et al., 2013*). Both loci contained multiple accessible regions, including elements open already in the naive *Mesp+* mesoderm (e.g. *Ebf_KhL24.35/36*), cardiopharyngeal-lineage-specific elements that open prior to gene activation but after induction of multipotent progenitors (e.g. *Ebf_KhL24.34*), and elements that open de novo in fate-restricted pharyngeal muscle precursors, where the gene is activated (e.g. *Ebf_KhL24.37*) (*Figure 5A,B*). Previous reporter gene expression assays identified the latter element, *Ebf_KhL24.37*, as a weak minimal enhancer with pharyngeal muscle-specific activity (*Wang et al., 2013*). CRISPR/Cas9-mediated mutagenesis assays followed by FISH indicated that each one of these elements is necessary for proper activation of *Ebf* in pharyngeal muscle progenitors (*Figure 5C–E*; *Figure 5—figure supplement 2B,C*). Consistently, we found that targeted deletions of individual accessible elements upstream of *Ebf* induced pharyngeal muscle precursor migration defects. When targeting *Ebf_KhL24.37*, 37 ± 5% (SE) of 28 hpf larvae showed such defects (n = 101, *Figure 5—figure supplement 3*). These observations indicate that each accessible *cis*-regulatory element upstream of *Ebf* is necessary for proper expression and subsequent pharyngeal muscle morphogenesis.

Consistent with the established roles of Hand-r, Tbx1/10 and Ets-mediated FGF-MAPK signaling in activating *Ebf*, the primed cardiopharyngeal-specific element (*KhL24.34*) contained Fox and bHLH motifs, and the more distal de novo-accessible minimal enhancer (*KhL24.37*) also contained putative Ets and RORγ binding sites, whereas the constitutively accessible elements (*KhC24.35* and *.36*) contained primarily CREB and T-box binding sites (*Figure 5C*; *Figure 5—figure supplement 2A*). *Tbx1/10* showed a similar logic, whereby a constitutively accessible upstream element (*KhC7.909*) acts as an enhancer of cardiopharyngeal expression (*Razy-Krajka et al., 2018*), and whose activity also requires a primed cardiopharyngeal-specific intronic element (*KhC7.914*) (*Figure 5—figure supplement 1A–F*). As a complement, and to more directly test the importance of enhancer accessibility, we targeted the intronic and distal elements in the *Tbx1/10* locus using dCas9::KRAB (*Klann et al., 2017*), which recruits deacetylases and presumably closes chromatin (*Sripathy et al., 2006*; *Groner et al., 2010*; *Schultz et al., 2002*; *Reynolds et al., 2012*; *Thakore et al., 2015*). We oserved loss of *Tbx1/10* expression and function, as evaluated by expression of its target, *Ebf* (*Wang et al., 2013*) (*Figure 5—figure supplement 4*).

Of note, *Ebf* expression is maintained by auto-regulation (*Razy-Krajka et al., 2018*), which requires separate intronic elements that harbor putative Ebf binding sites and open later (*Figure 5B*, *Figure 5—figure supplement 2A*). Together with Ebf's potent myogenic and anti-cardiogenic effects (*Razy-Krajka et al., 2014*; *Stolfi et al., 2014*; *Stolfi et al., 2010*; *Tolkin and Christiaen, 2016*), this auto-regulatory logic catalyzes the pharyngeal muscle fate, stressing the importance of spatially and temporally accurate onset of expression to avoid ectopic ASM specification at the expense of cardiac identities, especially in the second heart lineage. These observations suggest that pharyngeal muscle fate specification relies on 'combined enhancers', characterized by a combination of *trans*-acting inputs mediated by distinct elements with variable dynamics of accessibility, to control the onset of *Ebf* expression in the cardiopharyngeal mesoderm (*Figure 5F*).

To test whether 'combined enhancers' drive spatially and temporally accurate expression in pharyngeal muscle progenitors, we built a reporter containing multiple copies of the minimal, but weak, *Ebf* enhancer (*KhL24.37*) (*Wang et al., 2013*). Two and three copies of the *KhL24.37* element (2x and 3x *KhL24.37*) significantly increased reporter gene expression in pharyngeal muscle precursors (43 ± 3% SE for 2x *KhL24.37*; 58 ± 2% SE for 3x *KhL24.37*), compared to a single copy construct) (14 ± 3% SE for 1x *KhL24.37*) (*Figure 6A–C*), restoring reporter gene expression to levels similar to 'full length' upstream element encompassing all combined enhancers, with endogenous genomic spacing (Ebf-full length −3348 /- 178) (*Wang et al., 2013*) (80 ± 1%, SE). Remarkably, unlike the 'full length' combined enhancers, the 3x *KhL24.37* construct induced precocious reporter gene expression in the STVCs (89 ± 3%, SE) (n = 95, *Figure 6D–F*) causing an ectopic GFP expression in the second heart lineage (13 ± 2%, SE) (n = 218, *Figure 6B,C*). To test whether spacing between accessible elements could affect transcriptional output, we built a concatemer of *KhL24.37*, *.36*, *.35*, and *.34* elements without endogenous spacer sequences. This construct increased the proportion of embryos with ASM cells expressing the reporter to 92 ± 2% (SE, n = 130; *Figure 6C*), but it did not

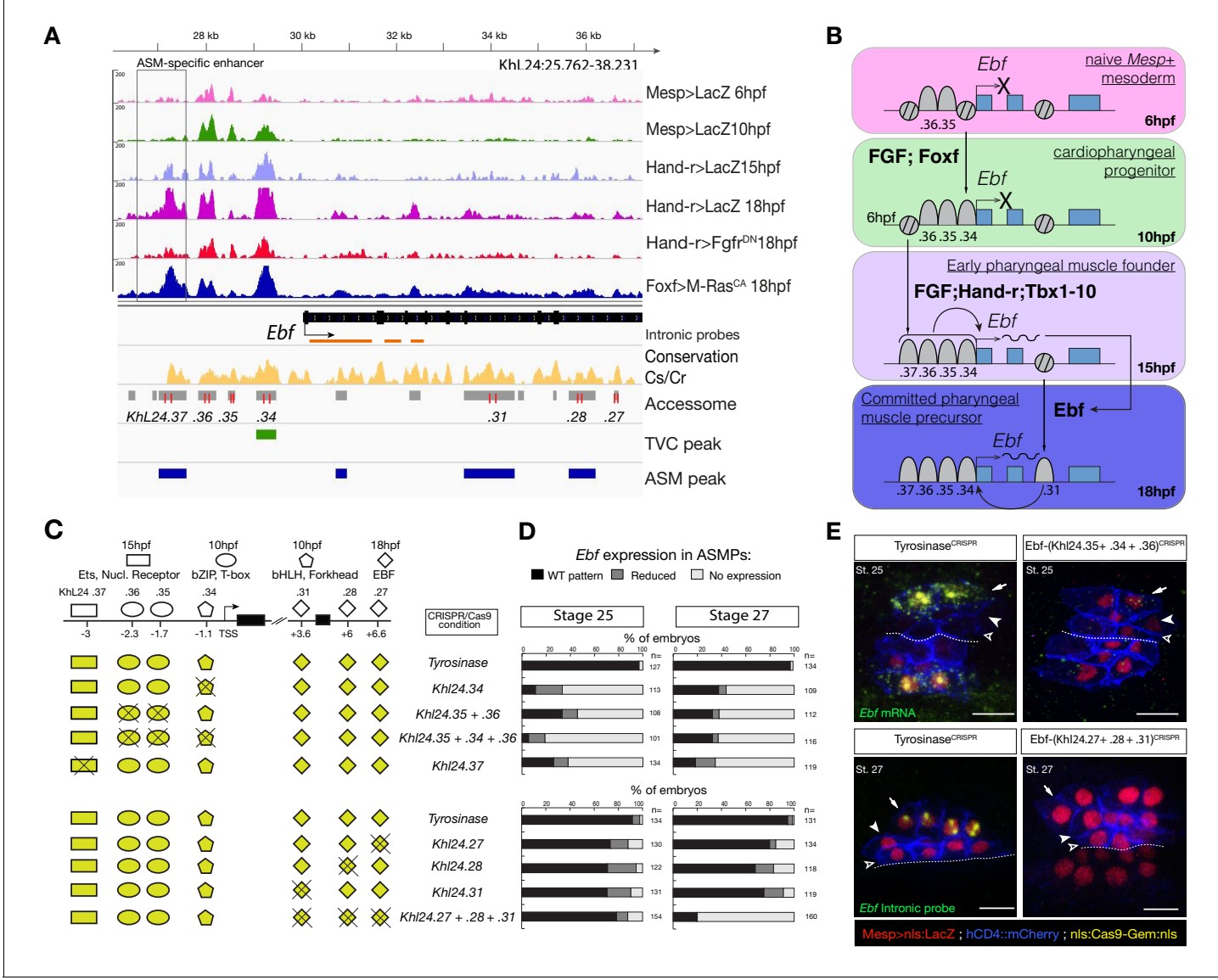

**Figure 5.** Combinations of *cis*-regulatory elements with distinct chromatin accessibility profiles are required for *Ebf* transcription in pharyngeal-muscle precursors. (A) A 12 kb region of the scaffold L24 displaying expression profiles of RNA-seq and chromatin accessibility profiles of ATAC-seq (normalized tag count) in the *Ebf* locus. sgRNAs used to target ATAC-seq peaks are shown in red; intronic antisense riboprobes are shown in orange (B) Schematic representation showing sequential opening of *cis*-regulatory elements required for *Ebf* activation in pharyngeal muscle founder cells, and maintenance by auto-regulation in committed precursor. (C) Schematic representation of *Ebf cis*-regulatory elements targeted for CRISPR/Cas9-mediated deletions. Shapes represent binding sites located in the regulatory elements and differentially accessible over time. (D) Proportions of larva halves showing the indicated *Ebf* transcription patterns, in indicated experimental conditions; all the treatments were significant versus *Tyrosinase* (Fisher exact test, $p < 0.001$). (E) Endogenous expression of *Ebf* visualized by in situ (green) in *Tyrosinase*[CRISPR] and upon CRISPR/Cas9-induced deletion of ATAC-seq peaks as indicated, at stage 25 (E) and 27 (F) based on *Hotta et al. (2007)*. For stage 25, an anti-sense riboprobe for the full length cDNA was used, whereas for stage 27 an intronic anti-sense riboprobe targeting the first three introns of *Ebf* transcript (orange lines) as previously used in *Wang et al. (2013)*. Nuclei of B7.5 lineage cells are labelled by *Mesp>nls::LacZ* and revealed with an anti beta-galactosidase antibody (red). *Mesp*-driven hCD4::mCherry accumulates at the cell membrane as revealed by anti mCherry antibody (Blue). Scale bar = 10 µm.

The online version of this article includes the following figure supplement(s) for figure 5:

**Figure supplement 1.** Combinations of *cis*-regulatory elements with distinct chromatin accessibility profiles are required for *Tbx1/10* transcription in pharyngeal-muscle precursors.

**Figure supplement 2.** *Ebf* regulatory regions showing differentially accessibility over time contain distinct binding motifs.

**Figure supplement 3.** CRISPR/Cas9-mediated deletions on individual accessible elements upstream of *Ebf* caused phenotypic impact on pharyngeal muscle precursors morphogenesis.

**Figure supplement 4.** Intronic and distal enhancer accessibility in the *Tbx1/10* locus tested by dCas9-KRAB.

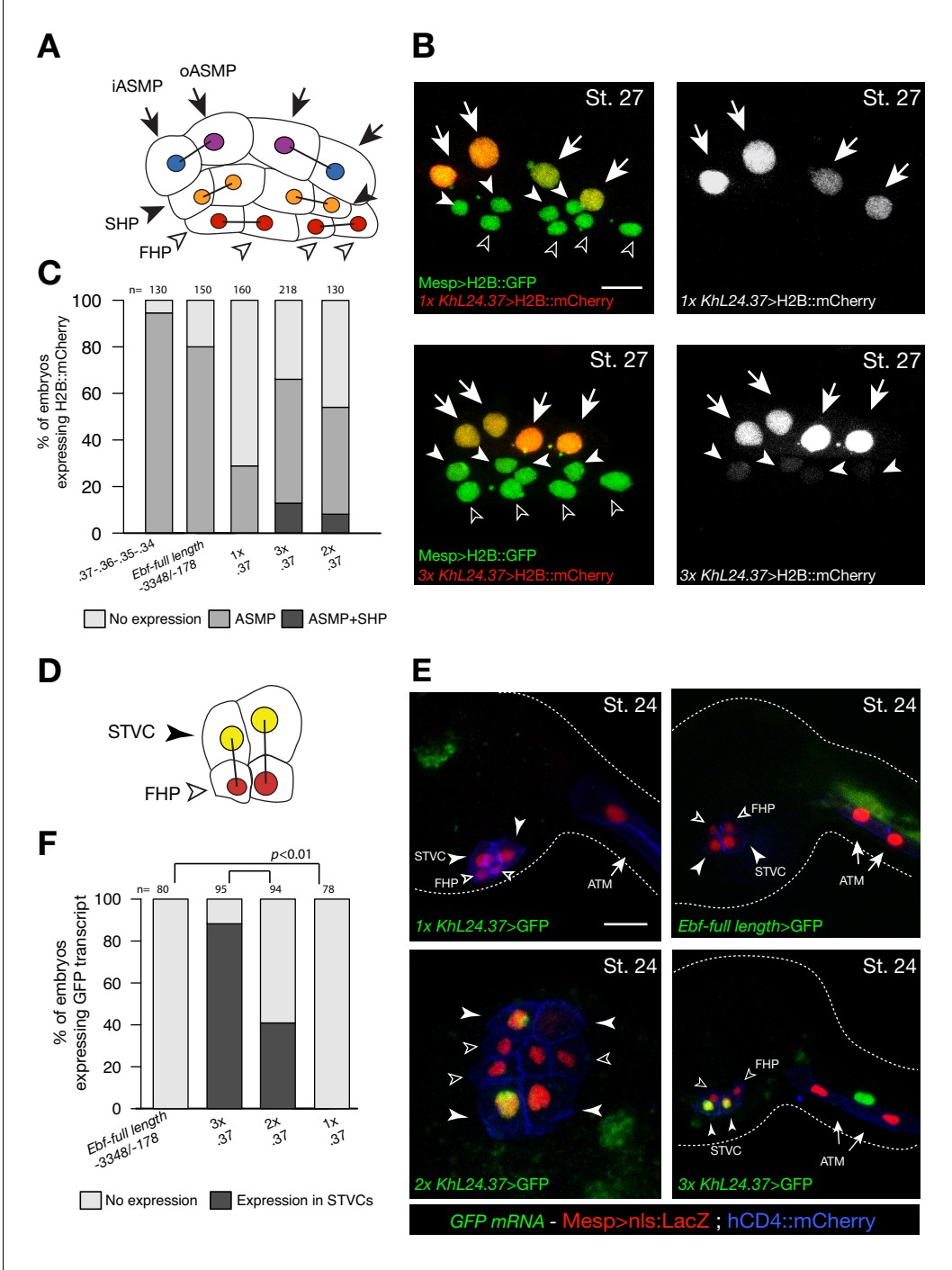

**Figure 6.** Multiple copies of a weak *Ebf* enhancer drive ectopic reporter gene expression. (**A**) Schematic representation of cardiopharyngeal lineage cells at Stage 27 (*Hotta et al., 2007*); First heart precursors (FHPs, red and open arrowheads), second heart precursors (SHPs, orange and arrows), inner ASM precursors and derivatives (iASMPs, violet and solid arrowhead), outer ASM precursors and derivatives (oASMPs, dark blue and solid arrowhead) (see (*Razy-Krajka et al., 2014*); black bars link sister cells. (**B**) Lineage tracing in individual larvae expressing single (1x) and multiple copy (3x) of *Ebf cis*-regulatory element, *KhL24.37*, driving H2B::mCherry (red) in cardiopharynegal progenitors at stage 27; B7.5 lineage is marked with *Mesp>H2B::GFP* (green). The single copy of *KhL24.37* element drives H2B::mCherry reporter expression specifically in the ASMPs (upper left panel, in white); three copies of *KhL24.37* (3x *L24.37*) drives expression in ASMPs and induces ectopic expression SHPs (lower left panel, in white). Experiment performed in biological replicate. Scale bar = 30 µm. (**C**) Proportions of embryos
*Figure 6 continued on next page*

Figure 6 continued

expressing H2B::mCherry in indicated cell-type progenitors by the indicated *cis*-regulaory elements. The 'full length' upstream region encompassing all combined enhancers with endogenous spacing (*Ebf*-full length −3348 /- 178) (*Wang et al., 2013*) as well as the concatemer of *KhL24.37, .36, .35*, and *.34* elements, lacking endogenous spacer sequences were used as controls. Statistical analysis using a Fisher exact test showed all comparisons with either control to be significant ($p < 0.01$); 'n' is the total number of individual halves scored per condition. (D) Schematic representation of cardiopharyngeal lineage cells at Stage 24 (*Hotta et al., 2007*); First heart precursors (FHPs, red and open arrowheads), secondary TVC (arrows). (E) Expression of GFP visualized by in situ hybridization on embryos at Stage 24 electroporated with single, multi copies and full-length of *Ebf cis*-regulatory element. Nuclei of B7.5 lineage cells are labelled by *Mesp>nls::LacZ* and revealed with an anti beta-galactosidase antibody (red). *Mesp*-driven hCD4::mCherry accumulates at the cell membrane as revealed by anti mCherry antibody (Blue). Scale bar = 10 µm. (F) Proportions of embryos showing GFP-driven by the indicated *Ebf cis*-regulatory element (Fisher exact test, $p < 0.01$).

induce ectopic expression in the second heart lineage, supporting the notion that combined enhancers drive high but spatially and temporally accurate expression.

We reasoned that high but precocious activation of *Ebf* should suffice to trigger the autoregulatory loop, and cause ectopic pharyngeal muscle specification in the cells that normally form the second heart lineage, as observed previously (*Razy-Krajka et al., 2014*; *Stolfi et al., 2010*). To directly test this possibility, we used different combinations of the *cis*-regulatory elements upstream of *Ebf* to drive expression of a functional *Ebf* cDNA, and assayed the effects endogenous *Ebf* expression and cardiopharyngeal fates (*Figure 7*). As expected, one copy of the *KhL24.37* element (1x *KhL24.37*) failed to cause ectopic activation of the endogenous locus, as evaluated using intronic probes, whereas using the *KhL24.37* trimer (3x *KhL24.37*) to drive expression of the *Ebf* cDNA sufficed to activate the endogenous locus ectopically in ~40% of the embryos, as shown by the presence of nascent transcripts in 4 out of 6 nuclei per side, instead of 2 (*Figure 7A,B*). This observation is consistent with our model that the upstream elements mediate high activating inputs for the onset of *Ebf* expression, whereas maintenance upon commitment to a pharyngeal muscle identity relies on autoregulation (*Figure 5F*; *Razy-Krajka et al., 2018*).

These results suggested that inaccurate activation and maintenance of *Ebf* expression would cause ectopic pharyngeal muscle specification at the expense of the second heart lineage. We tested this possibility by analyzing pharyngeal muscle morphogenesis in stage 30 larvae, which is characterized by collective migration away from heart progenitors and formation of a ring of atrial siphon muscle precursors (*Figure 7*). We used the STVC-specific *Tbx1/10* enhancer to visualize both the pharyngeal muscle precursors, which migrate and form a ring, and the second heart precursors, which remain associated with the *Tbx1/10*-negative first heart lineage (*Figure 7D*; *Razy-Krajka et al., 2018*). Remarkably, *Ebf* misexpression using three copies of the *KhL24.37* element induced cells that normally form the second heart lineage to migrate alongside the pharyngeal muscles in 38% of larvae at 28 hpf (38 ± 3%, SE) (n = 265, *Figure 7C–D*). Importantly, neither one copy of *KhL24.37* (1x *KhL24.37*), nor the full combined enhancers, with or without endogenous spacers, sufficed to cause substantial fate transformation of *Tbx1/10+* second heart lineage into migratory pharyngeal muscle precursors (*Figure 7C–D*). These results indicate that driving expression of an *Ebf* cDNA by multimerizing a weak *Ebf* enhancer sufficed to cause ectopic activation of the endogenous locus and transformation of second heart lineage cells into migratory pharyngeal muscle precursors.

Taken together, these results provide evidence for a decoupling between two essential aspects of transcriptional activation, whereby multiple copies of a single regulatory element enable robust gene expression, compatible with precise fate specification, albeit at the expense of spatial and temporal accuracy; whereas individual regulatory elements appear to integrate distinct *trans*-acting inputs controlling enhancer accessibility and/or activity, thus increasing the repertoire of regulatory inputs controlling developmental gene expression (*Figure 7E*). In other words, while multiple elements are required for proper activation of cell fate determinants, such as *Tbx1/10* and *Ebf*, in a manner reminiscent of super- and shadow enhancers (*Lagha et al., 2012*), we propose that combined enhancers foster spatially and temporally accurate cell fate decisions.

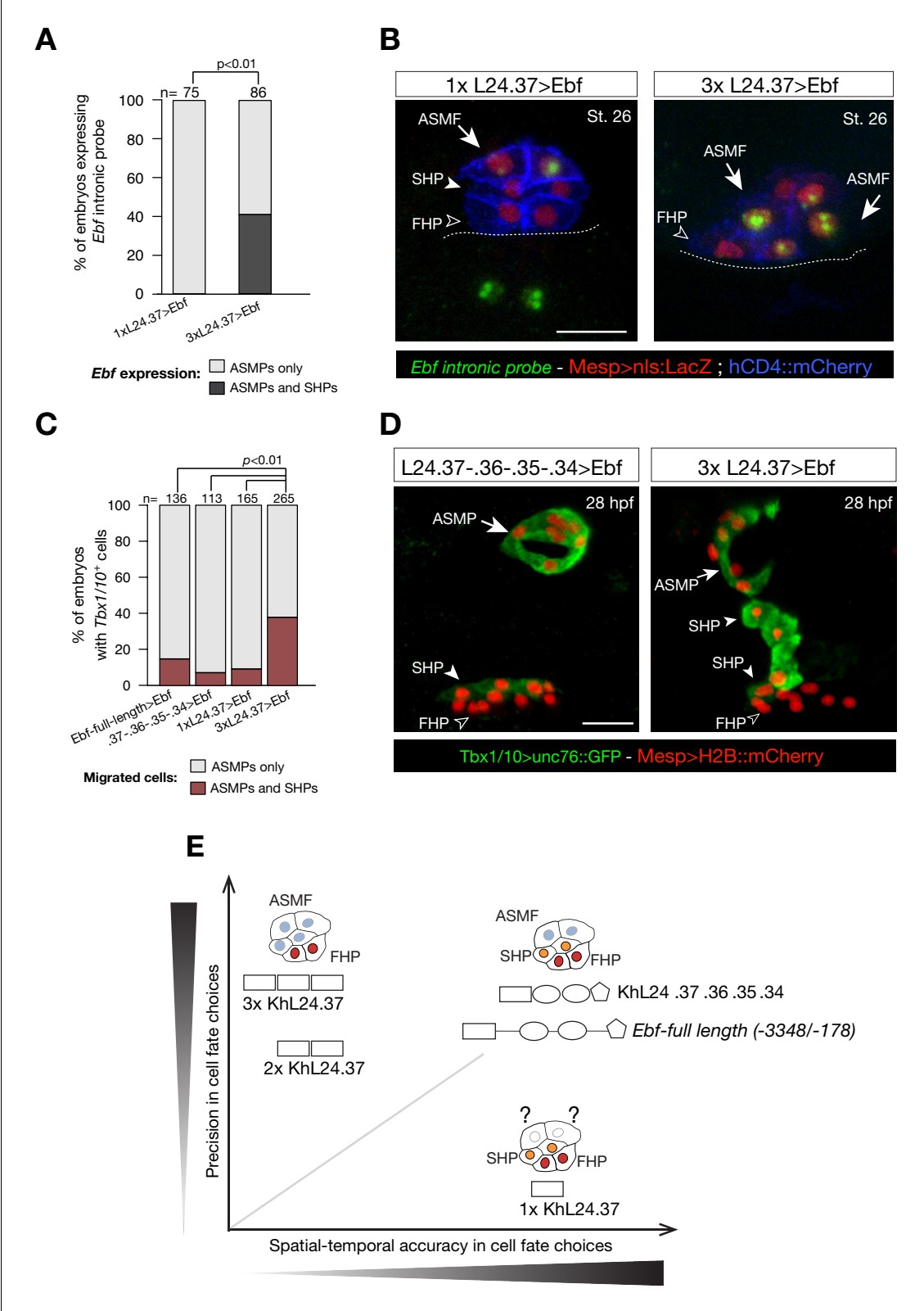

**Figure 7.** Multimer of one weak *Ebf* enhancer drives ectopic pharyngeal muscle fate specification. (**A**) Targeted expression of an *Ebf* cDNA by three copies of *KhL24.37* element induces expression of endogenous *Ebf* in four cells compared to the control where *Ebf* is detected only in the two ASMPs. Proportions of embryos expressing *Ebf* in ASMPs only or in ASMPs and SHPs in the indicated conditions. The single *KhL24.37* cis-regulatory was used as control. (**B**) Expression of *Ebf* visualized by in situ hybridization on embryos at Stage 26 electroporated with single or three copies *KhL24.37* element

*Figure 7 continued on next page*

*Figure 7 continued*

driving *Ebf* cDNA. Intron-specific probes show nascent *Ebf* transcripts in ASMPs. Nuclei of B7.5 lineage cells are labelled by *Mesp>nls::LacZ* and revealed with an anti beta-galactosidase antibody (Red). *Mesp*-driven hCD4::mCherry accumulates at the cell membrane as revealed by anti mCherry antibody (Blue). Scale bar = 10 μm. (C) Proportions of embryos showing GFP-driven STVC-specific enhancer of *Tbx1/10* (Fisher exact test, *p* < 0.01). The 'full length' upstream element encompassing all combined enhancers with endogenous spacing (Ebf-full length −3348 / -178) as well as the 'full length' without endogenous spacing (*KhL24.37, .36, .35, .34*) and the single copy *KhL24.37* element driving *Ebf* were used as a positive controls. (D) Example of an embryo at 28 hpf showing GFP expression only in the ASMP (solid arrowhead) and SHP (arrow) but not in the FHP (open arrowheads), where *Tbx1/10* enhancer is not active. Targeted expression of *Ebf* cDNA by three copies of *KhL24.37* element induces cells that normally form the second heart lineage to migrate alongside the pharyngeal muscle. Nuclei of B7.5 lineage cells are labelled by *Mesp>H2B::mCherry*. Scale bar = 15 μm. (E) Proposed role of combinatorial logics in fostering both precision and spatial and temporal accuracy of cell fate choices. The *Ebf*-full length, with or without endogenous spacers, fosters spatial/temporal accurate and precise cell fate choice, whereas the single copy of *KhL24.37* (1x *L24.37*) gives spatiotemporally accurate reporter expression but is likely insufficient to induce a precise ASM fate. Multiple copies of *KhL24.37* *cis*-regulatory element rescue a high reporter activity that reflects precise pharyngeal fate at the expense of spatial and temporal accuracy. The shapes of the distinct *cis*-regulatory elements are as in *Figure 5C*. Statistical analysis using a Fisher exact test (*p* < 0.01); 'n' is the total number of individual halves scored per condition.

## Discussion

We characterized the accessible genome of the tunicate *Ciona*, with a special focus on the cardio-pharyngeal lineage that produces heart and pharyngeal muscles. As seen in other systems, less than 10% of the *Ciona* genome is accessible, and distributed across thousands of short regions, most of which are stably accessible across time and lineages, especially promoter regions. By contrast, developmentally regulated regions either closed upon induction of multipotent progenitors or opened specifically in the cardiopharyngeal lineage in response to FGF-MAPK signaling and Foxf activity. The latter elements were predominantly found in intergenic and intronic regions, and near cardio-pharyngeal markers, consistent with their function as transcriptional enhancers. Similarly to other Forkhead factors (*Zaret and Carroll, 2011*), *Ciona* Foxf is required to open cardiopharyngeal elements for either immediate or later activation in multipotent or fate-restricted progenitors, respectively. Notably, Foxf homologs play deeply conserved roles in visceral muscles specification (*Jakobsen et al., 2007*; *Scimone et al., 2018*; *Zaffran et al., 2001*), including during heart development in mammals (*Hoffmann et al., 2014*). GATA motifs are also over-represented among cardio-pharyngeal-specific elements, consistent with a conserved role for GATA homologs in heart development (*Holtzinger and Evans, 2007*; *Molkentin et al., 1997*; *Qian and Bodmer, 2009*; *Reiter et al., 1999*; *Sorrentino et al., 2005*; *Zhao et al., 2008*). As combinations of FOX and GATA inputs play well-established roles in early endoderm specification (*Cirillo et al., 2002*), we speculate that cardiopharyngeal regulatory programs were built upon an ancestral endomesodermal chromatin landscape during Olfactores evolution.

The majority of cell-type-specific markers expressed de novo are associated with 'primed accessible' elements, as observed in numerous systems including cardiac differentiation of embryonic stem cells (*Paige et al., 2012*; *Wamstad et al., 2012*), and consistent with the role of pioneer factors in establishing competence for subsequent activation (*Zaret and Carroll, 2011*). In the case of *Tbx1/10* and *Ebf*, spatially and temporally accurate activation is essential to permit the emergence of first and second cardiac, and pharyngeal muscle lineages (*Razy-Krajka et al., 2018*; *Razy-Krajka et al., 2014*; *Wang et al., 2019*; *Wang et al., 2013*). We found that several elements, exhibiting distinct accessibility dynamics, are required for proper activation of both *Tbx1/10* and *Ebf*. Specifically, a minimal distal enhancer proved necessary, but not sufficient for *Ebf* activation in newborn pharyngeal muscle precursors. By contrast, multiple copies of the same element sufficed to restore high transcriptional activity, but caused precocious activation in the *Tbx1/10+* multipotent progenitors (aka STVCs, *Figure 1A* and *Figure 6E*), ectopic GFP expression in the second heart lineage (*Figure 6*), and eventually heart-to-pharyngeal muscle fate transformation when used to express *Ebf* itself (*Figure 7*). We propose that, whereas the activity of multiple elements with similar spatio-temporal transcriptional outputs permits precise and robust gene activation (*Bentovim et al., 2017*; *Lagha et al., 2012*), the modular organization of combined enhancers increases the repertoire of regulatory inputs, acting through both accessibility and activity, to control gene activation. Together with canalizing mechanisms, including positive auto-regulatory feedbacks, such multi-level

combinatorial inputs achieve exquisite spatio-temporal control while permitting strong activation, thus ensuring both precise and accurate developmental fate choices (*Figure 7E*).

## Materials and methods

### Animals and electroporations

Gravid wild *Ciona intestinalis* type A, now called *Ciona robusta* (*Pennati et al., 2015*), were obtained from M-REP (Carlsbad, CA, USA), and kept under constant light to avoid spawning. Gametes from several animals were collected separately for in vitro cross-fertilization followed by dechorionation and electroporation as previously described (*Christiaen et al., 2009a*) , and cultured in filtered artificial seawater (FASW) in agarose-coated plastic Petri dishes at 18°C. Different quantities of plasmids were electroporated depending on the constructs: the amount of fluorescent reporter DNA ( *Mesp>nls::lacZ* , *Mesp>hCD4::mCherry* , *Mesp>tagRFP* , *MyoD905>eGFP* and *Hand-r>tagBFP*) and *NLS::lacZ* was typically 50 µg, but only 15 µg for *Mesp>H2B::mCherry*. For perturbation constructs (*Mesp>Fgfr$^{DN}$*, *Mesp>Mek$^{S216D,S220E}$*, *Foxf>M Ras$^{CA}$*, *Hand-r>Fgfr$^{DN}$*), 70 µg were usually electroporated, except for *Mesp>nls::Cas9-Gem::nls* (30 µg) and pairs of *U6>sgRNA* plasmids (25 µg each).

### Molecular cloning

Putative enhancers were amplified from *Ciona robusta* genome using primers containing specific sequence tails (*Supplementary file 4*) for subcloning into a vector upstream of a basal promoter from *Zfpm* (aka *Friend of GATA*/*Fog* [*Rothbächer et al., 2007*]) driving expression of green fluorescent protein (GFP) fused to cytoplasmic unc76 (*Stolfi et al., 2010*).

The 'full length' *Ebf* enhancer with non-endogenous spacing in the genome (*KhL24.37*, .36, .35, .34) was generated by synthesizing DNA fragment (Twist Bioscience) and subcloning into the full-length reporter plasmids. *nls::dCas9-KRAB::nls* was derived from 'pLV-dCas9-KRAB-PGK-HygR' (*Klann et al., 2017*) (Addgene plasmid: #83890) and inserted downstream of the promoter *Mesp* (*Davidson et al., 2005*).

### CRISPR/Cas9-mediated mutagenesis of ATAC-seq peaks

Two to four single guide RNAs (sgRNA) with Doench scores (http://crispor.tefor.net; *Haeussler et al., 2016*) higher than 60 were designed to induce deletions in selected accessible elements using CRISPR/Cas9 in the B7.5 lineage as described (*Gandhi et al., 2017*). sgRNAs targeting non-overlapping sequences per gene are listed in *Supplementary file 5*. The efficiency of sgRNAs was evaluated using the peakshift method as described (*Gandhi et al., 2017*) (*Figure 2—figure supplement 3B–C*; *Figure 2—figure supplement 6C*). CRISPR/Cas9-mediated deletions were also evaluated by PCR-amplification directly from embryo lysates following electroporated with *Eef1a>nls::Cas9-Gem::nls* (*Figure 2—figure supplement 6C*). sgRNAs were expressed using the *Ciona robusta U6* promoter (*Stolfi et al., 2014*) (*Figure 2—figure supplement 3D*; *Figure 2—figure supplement 6C–D*). For each peak, three or four guide RNAs were used in combination with 25 µg of each expression plasmid. 25 µg of *Mesp>nls::Cas9-Gem::nls* and *Mesp>nls::dCas9-KRAB::nls* plasmids were co-electroporated with guide RNA expression plasmids for B7.5 lineage-specific CRISPR/Cas9-mediated mutagenesis. Two guide RNAs were used to mutagenize *Tyrosinase*, which is not expressed in the cardiopharyngeal lineage and thus used to control the specificity of the CRISPR/Cas9 system (*Wang et al., 2019*).

### Fluorescent in situ Hybridization-Immunohistochemistry (FISH-IHC) in *Ciona* embryos

FISH-IHC were performed as previously described (*Christiaen et al., 2009b*; *Razy-Krajka et al., 2018*) . Embryos were harvested and fixed at desired developmental stages for 2 hr in 4% MEM-PFA and stored in 75% ethanol at −20°C. Antisense RNA probes were synthesized as described (*Racioppi et al., 2014*). For *Ebf* FISH-IHC, an anti-sense riboprobe targeting the full length cDNA was used at Stage 24 (*Figure 5—figure supplement 2B*) and 25 (*Figure 5E*, upper panel) where an anti-sense riboprobe targeting the first three intronic elements in the *Ebf* transcript as previously used in *Wang et al. (2013)* was used for stage 27 (*Figure 5E*, lower panel; *Figure 6E*). The template

for *Fbln* anti-sense riboprobe was PCR-amplified with the following oligos: Fbln_pb_fw (5' TTGCGC TAAGTCATGACAGC 3'), Fbln_pb_rev (5'CATTTGCCGATTCAGCTATGT3'). In vitro antisense RNA synthesis was performed using T7 RNA Polymerase (Roche, Cat. No. 10881767001) and DIG RNA Labeling Mix (Roche, Cat. No. 11277073910). Anti-Digoxigenin-POD Fab fragment (Roche, IN) was first used to detect the hybridized probes, then the signal was revealed using Tyramide Signal Amplification (TSA) with Fluorescein TSA Plus Evaluation Kits (Perkin Elmer, MA). Anti–β-galactosidase monoclonal mouse antibody (Promega) was co-incubated with anti-mCherry polyclonal rabbit antibody (Bio Vision, Cat. No. 5993–100) for immunodetection of *Mesp>nls::lacZ* and *Mesp>hCD4:: mCherry* products respectively. Goat anti-mouse secondary antibodies coupled with AlexaFluor-555 and AlexaFluor-633 were used to detect β-galactosidase-bound mouse antibodies and mCherry-bound rabbit antibodies after the TSA reaction. FISH samples were mounted in ProLong Gold Anti-fade Mountant (ThermoFisher Scientific, Waltham, MA. Catalog number P36930).

## Imaging

Images were acquired with an inverted Leica TCS SP8 X confocal microscope, using HC PL APO $63\times/1.30$ objective. Z-stacks were acquired with 1 μm z-steps. Maximum projections were processed with maximum projection tools from the LEICA software LAS-AF.

## Cell dissociation and FACS

Sample dissociation and FACS were performed as previously described (*Christiaen et al., 2016*; *Wang et al., 2018*). Embryos and larvae were harvested at 6, 10, 15, 18 and 20 hpf in 5 ml borosilicate glass tubes (Fisher Scientific, Waltham, MA. Cat.No. 14-961-26) and washed with 2 ml calcium- and magnesium-free artificial seawater (CMF-ASW: 449 mM NaCl, 33 mM $Na_2SO_4$, 9 mM KCl, 2.15 mM $NaHCO_3$, 10 mM Tris-Cl pH 8.2, 2.5 mM EGTA). Embryos and larvae were dissociated in 2 ml 0.2% trypsin (w/v, Sigma, T- 4799) in CMF-ASW by pipetting with glass Pasteur pipettes. The dissociation was stopped by adding 2 ml filtered ice cold 0.05% BSA CMF-ASW. Dissociated cells were passed through 40 μm cell-strainer and collected in 5 ml polystyrene round-bottom tube (Corning Life Sciences, Oneonta, New York. REF 352235). Cells were collected by centrifugation at 800 g for 3 min at 4°C, followed by two washes with ice cold 0.05% BSA CMF-ASW. Cell suspensions were filtered again through a 40 μm cell-strainer and stored on ice. Cell suspensions were used for sorting within 1 hr. B7.5 lineage cells were labeled by *Mesp>tagRFP* reporter. The B-line mesenchyme cells were counter-selected using *MyoD905>GFP* as described (*Christiaen et al., 2016*; *Wang et al., 2018*). The TVC-specific *Hand-r>tagBFP* reporter was used in a 3-color FACS scheme for positive co-selection of TVC-derived cells (*Wang et al., 2018*), in order to minimize the effects of mosaicism. Dissociated cells were loaded in a BD FACS AriaT M cell sorter. 488 nm laser, FITC filter was used for GFP; 407 nm laser, 561 nm laser, DsRed filter was used for tagRFP and Pacific BlueTM filter was used for tagBFP. The nozzle size was 100 μm. eGFP$^+$ mesenchyme cells were collected for downstream ATAC-seq analysis, whereas tagRFP$^+$, tagBFP$^+$ and eGFP$^-$ cardiopharyngeal lineage cells were collected for both ATAC- and RNA-seq analyses.

## Preparation and sequencing of ATAC-seq library

ATAC-seq was performed with minor modifications to the published protocol (*Buenrostro et al., 2015*). 4,000 cells obtained by FACS were centrifuged at 800 x *g* for 4 min at 4°C for each sample. Cells were resuspended in 5 μL of cold lysis buffer (10 mM Tris-HCl [pH 7.4], 10 nM NaCl, 3 nM MgCl$_2$, 0,1% v/v Igepal CA-360 [Sigma-Aldrich] and incubated on ice for 5 min. After centrifugation of the cells at 500 x *g* for 10 min at 4°C, the supernatant was discarded. The tagmentation reaction was performed at 37°C for 30 min using Nextera Sample Preparation kit (Illumina) with the addition of a tagmentation-stop step by the addition of EDTA to a final concentration of 50 nM and incubation at 50°C for 30 min (*Hockman et al., 2019*). After tagmentation, transposed DNA fragments were amplified using the following PCR condition: 1 cycle of 72°C for 5 min and 98°C for 30 s, followed by 12 to 14 cycles of 98°C for 10 s, 63°C for 30 s and 72°C for 1 min. Amplified libraries were purified using PCR purification MinElute kit (Qiagen), and the quality of of the purified library was assessed by using 2100 Bioanalyzer (Agilent Technologies) and a High Sensitivity DNA analysis Kit (Agilent Technologies) to confirm a period pattern in the size of the amplified DNA. After size selection by using Agencourt Ampure XP beads (Beckman Coulter, Cat#A633880) with a bead-to-sample

ratio of 1.8:1, the libraries were sequenced as paired-end 50 bp by using the HiSeq 2500 platform (Illumina) according to the manufacturer's instructions. An input control was also generated by using 10 ng of *C. robusta* genomic DNA.

## ATAC-seq data analysis

### Alignment of ATAC-seq reads

Raw reads were preprocessed by FastQC (version 0.11.2, http://www.bioinformatics.babraham.ac.uk/projects/fastqc) and adaptors were trimmed using Trim Galore (version 0.4.4, http://www.bioinformatics.babraham.ac.uk/projects/trim_galore) before being aligned to *Ciona robusta* genome (joined scaffold (KH), http://ghost.zool.kyoto-u.ac.jp/datas/JoinedScaffold.zip) using Bowtie2 (version 2.3.2, *Langmead and Salzberg, 2012*) with the parameters `-no-discordant -p 20 -X 1000`. Reads with mapping quality score > 30 were kept for downstream analysis using SAMtools (version 1.2, *Li et al., 2009*). Mitochondrial reads were removed using NGSutils 0.5.9 (*Breese and Liu, 2013*). At least three replicates for each sample were merged for peak calling by MACS2 (version 2.7.9) (*Zhang et al., 2008*) (`-nomodel -nolambda -gsize 1.2e8`). To correct for nonspecific sequencing biases, we subtracted sequenced gDNA from these libraries (*Toenhake et al., 2018*). We defined the accessome by concatenating all narrow peaks from MACS2 using bedtools merge (2.26.0) (*Quinlan and Hall, 2010*). Peaks were filtered for length > 50 bp, to obtain a final number of 56,090 peaks.

### Accessome annotation

We annotated each peak to all transcripts from the *C. robusta* genome (http://ghost.zool.kyoto-u.ac.jp/datas/KH.KHGene.2013.gff3.zip) within 10 kb using GenomicRanges (version 1.28.6, *Lawrence et al., 2013*). We also identified whether each peak overlapped with the genomic feature categories 'TSS, 5' UTR, intron, exon, 3' UTR, TTS, or intergenic region' (*Figure 1—figure supplement 1C*). The KH2013 gene IDs are identical to the KH2012 gene IDs after merging all transcripts of each gene. Only the genomic feaatures differ between the two annotations. We used a TSS-seq dataset of high-density TSS regions, or TSS clusters (TSCs) (*Yokomori et al., 2016*) from *Ciona* larvae as guidelines for core promoter length. We used the mean TSC length plus two standard deviations to define a window of 107 bp upstream and downstream of the start of the 5' UTR of any transcript as the TSS. We defined a window of 1 kb upstream from the TSS to be the promoter, which we further divided into two 500 bp windows. We defined the TTS to be a window 200 bp upstream and downstream from the end of the 3' UTR of any transcript. We defined regions not covered by any feature to be intergenic. We performed a two-tailed binomial test for enrichment of accessible elements in each of these features, as well as the previously published TSCs. The fraction of the genome covered by each feature was used as the null probability. We tested for differential expression of promoters by extending the TSS 1 kb upstream, then aligning bulk RNA-seq reads to this putative promoter region.

### Differential accessibility analysis

We evaluated the significant change in chromatin accessibility between different samples using DESeq2 (*Love et al., 2014*) with the parameters (method = 'LRT', alpha = 0.05, cooksCutoff = FALSE) for differential accessibility analysis. Libraries with fewer than 500,000 reads were removed. We performed a likelihood ratio test for four models:

1. the 10 hpf model, tested for difference in accessibility between any of the conditions at 10 hpf (*Mesp>Fgfr$^{DN}$*, *Mesp>Mek$^{S216D,S220E}$*, *U6>Foxf.1*, *U6>Foxf.4*, listed in *Supplementary file 5*) and controls (*Mesp>nls::LacZ* and *U6>Ngn.p1*, *U6>Ngn.p2* used for *Foxf$^{CRISPR}$*, listed in *Supplementary file 5*).

2. the 15–20 hpf model, tested for difference in accessibility between *Hand-r>Fgfr$^{DN}$* and control *Hand-r>LacZ* at any of the later time points (15, 18 or 20 hpf), *Foxf>M Ras$^{CA}$* and *Foxf>nls::LacZ* for the 18 hpf samples, versus a null model where accessibility is only dependent on time.

3. the time model, tested for any change in accessibility between control B7.5 samples (*Mesp>nls::LacZ* and *Hand-r>nls::LacZ*) at 6, 10, 15, 18 and 20 hpf versus a null model dependent where accessibility is not changing.

4. the control vs. mesenchyme model, tested for any difference in accessibility between B7.5 control (*Mesp>nls::LacZ* and *Hand-r>nls::LacZ*) and mesenchymal (*MyoD905>GFP*) samples versus a null model dependent at 10, 15, 18 and 20 hpf.

## Cell-type-specific accessibility

We defined cell-type-specific accessibility based on the peak sets:

- Elements opened by FGF-MAPK at 10 hpf: the union of peaks closed in *Mesp>Fgfr* vs. *Mesp>nls::LacZ* ($\log_2$(FC) < −0.5 and FDR < 0.05) and peaks open in *Mesp>Mek$^{S216D,S220E}$* vs. *Mesp>Fgfr$^{DN}$* ($\log_2$(FC)> 0.5 and FDR < 0.05) in model 1.
- Elements closed in FGF-MAPK at 10 hpf: the union of peaks closed in *Mesp>Mek$^{S216D,S220E}$* vs. *Mesp>nls::LacZ* and peaks closed in *Mesp>Mek$^{S216D,S220E}$* vs. *Mesp>Fgfr$^{DN}$* ($\log_2$(FC) < −0.5 and FDR < 0.05) in model 1.
- Elements opened in FGF-MAPK at 18 hpf: the union of peaks closed in *Hand-r>Fgfr$^{DN}$* vs. *Hand-r>nls::LacZ* ($\log_2$(FC) < −0.5 and FDR < 0.05 in the 15-20 hpf model), open in *Foxf>M Ras$^{CA}$* vs. *Foxf>nls::LacZ*, and open in *Foxf>M Ras$^{CA}$* vs. *Hand-r>Fgfr$^{DN}$* ($\log_2$(FC) > 0.5 and FDR < 0.05) in model 2.
- Elements closed in FGF-MAPK at 18 hpf: the union of peaks open in *Hand-r>Fgfr$^{DN}$* vs. *Hand-r>nls::LacZ* ($\log_2$(FC) > 0.5 and FDR < 0.05 in the 15–20 hpf model), closed in *Foxf>M Ras$^{CA}$* vs. *Foxf>nls::LacZ*, and closed in *Foxf>M Ras$^{CA}$* vs. *Hand-r>Fgfr$^{DN}$* ($\log_2$(FC) < −0.5 and FDR < 0.05) in model 2.
- De novo elements: peaks closed in control 10 vs. 18 hpf ($\log_2$(FC) < −0.5 and FDR < 0.05 in model three that are not opened by FGF-MAPK at 10 hpf.
- Primed elements: peaks opened in FGF-MAPK at 10 hpf that are not closed in *Mesp>LacZ* 10 vs. 18 hpf.

We tested for enrichment of peaks overlapping with each category of genomic feature in each of these peak sets (*Figure 2—figure supplement 2C*).

## Definition of gene sets

We used cardiopharyngeal lineage cell-type-specific gene sets (cardiopharyngeal markers), including primed and de novo-expressed Cardiac and ASM/pharyngeal muscle as well as secondary TVC (STVC), first heart precursor (FHP), second heart precursor (SHP) markers defined by scRNA-seq (*Wang et al., 2019*). Anterior tail muscles (ATM) marker genes were derived from publicly available expression data (*Brozovic et al., 2018*) (*Supplementary file 6*). To these we added the gene sets either activated or inhibited by MAPK at 10 or 18 hpf as defined by microarray (*Christiaen et al., 2008*) and Bulk RNA-seq analysis (*Wang et al., 2019*).

## Gene set enrichment analysis (GSEA)

We performed GSEA on elements associated with these gene sets using fgsea 1.2.1 (*Sergushichev, 2016*) with parameters: minSize = 5, maxSize = Inf, nperm = 10000. Because the gene sets used for GSEA need not be exclusive, we can use ATAC-seq peaks as 'genes' and define a gene set as all peaks annotated to genes in the set. We measured enrichment for peaks associated with the gene sets in open peaks (indicated by a positive enrichment score, ES) or closed peaks (indicated by a negative ES) in each comparison.

For GSEA of the ATAC-seq data, peaks were ranked by $\log_2$(FC) for a specific pairwise comparison, and all peaks annotated to a gene in a given gene set (as described above) were considered members of that gene set. The normalized enrichment score (NES) is a measure of whether members of the gene set are enriched at the top or bottom of a ranked list (*Figure 1D*). NES is calculated by keeping a running tally of whether or not the peak at each position of the list is a member of the gene set and comparing the maximum value to that of a null distribution calculated from a random permutation of peaks. For example, In *Figure 1D,* a positive NES indicates that peaks associated with a gene set are more open at 6 hpf than at 10 hpf, while a negative NES indicates that peaks associated with a gene set are more accessible at 10 hpf than at 6 hpf.

### *Ciona robusta* motif inference

We obtained inferred *C. intestinalis* (former name of *C. robusta*) transcription factor (TF) binding motifs from CIS-BP (*Weirauch et al., 2014*). The TF binding motifs were inferred based on similarity of DNA-binding domain sequences between *C. robusta* TFs and those of TFs with known binding motifs. A strong correlation has been observed between conserved DNA-binding domains and bound motifs (*Weirauch et al., 2014*).

## Motif selection

To the inferred motifs we added SELEX-seq motifs for *C. robusta* from *Nitta et al. (2019)*. We then searched the *C. robusta* genome for orthologs to known motifs from the HOMER database (*Heinz et al., 2010*). All motifs with orthologous *C. robusta* TFs were added to the set of candidate TF motifs. This set of candidate TF motifs consisted of 1,745 motifs associated with 294 *C. robusta* TFs. Of these, 1,228 motifs could be unambiguously associated with one of 237 TFs. To obtain motifs for the remaining 57 TFs, to minimize redundancy, we remove from consideration all associations to TFs with an assigned motif, then assign all unambiguous motifs to a TF. This iteration continues until no more unambiguous motifs can be assigned. Motifs for the remaining TFs are assigned by repeating the process for motifs associated with only two TFs, then three TFs, until each TF has an assigned motif. Our final set of motifs had 1,487 members, with the median TF associated with two motifs.

For *Figure 1—figure supplement 2E*, we used previously published eukaryotic core promoter motifs (*Haberle and Stark, 2018*). For *Figure 1—figure supplement 2F*, correlations were performed on the inferred *C. robusta* DNA-binding motifs.

## Motif analyses of ATAC-seq

We compiled the *C. robusta* genome (http://ghost.zool.kyoto-u.ac.jp/datas/JoinedScaffold.zip) as a BSGenome (version 1.44.2) package using 'forgeBSgenomeDataPkg'. We searched for all motifs present in accessible elements using motifmatchr (version 0.99.8) and TFBSTools (version 1.14.2, *Tan and Lenhard, 2016*). Background nucleotide frequencies were computed for the whole accessome. Matches were returned if they passed a cutoff of $p < 5*10^{-5}$. This cutoff was used to set a score threshold using the standard method of comparing the probability of observing the sequence given the PWM versus the probability of observing the sequence given the background nucleotide sequence of the accessible elements (*Staden, 1989*). We tested for motif enrichment in a peak set versus the accessome using a one-tailed hypergeometric test. The odds ratio represents the probability that an outcome (a peak containing the indicated motif) will occur if the peak is contained in the indicated peak set, compared to the probability of the outcome occurring in an element randomly selected from the accessome. For *Figure 4—figure supplement 4A*, the test indicates the probability that an element will contain a motif given when the element is accessible.

Primed or de novo accessible elements associated with de novo cardiac expressed genes were considered cardiac accessible if they had $\log_2(FC) < 0$ in *Foxf>M Ras$^{CA}$* vs. *Foxf>nls::LacZ* or $\log_2(FC) > 0$ in *Hand-r>Fgfr$^{DN}$* vs. *Hand-r>nls::LacZ*. Primed or de novo elements associated with de novo pharyngeal muscle (ASM) genes were considered ASM accessible if they had $\log_2(FC) > 0$ in *Foxf>M Ras$^{CA}$* vs. *Foxf>nls::LacZ* or $\log_2(FC) < 0$ in *Hand-r>Fgfr$^{DN}$* vs. *Hand-r>nls::LacZ*. The peak sets tested were primed cardiac accessible peaks associated with de novo cardiac expressed genes, de novo cardiac accessible peaks associated with de novo cardiac expressed genes, primed ASM accessible peaks associated with de novo ASM expressed genes, and de novo ASM accessible peaks associated with de novo ASM expressed genes. To ensure that the elements' accessibility is not reflecting expression of primed genes, elements were removed if they were also associated with a gene expressed earlier in the cardiopharyngeal lineage. Elements associated with de novo ASM genes were removed if they were also associated with a cardiac or FGF-MAPK inhibited at 18 hpf gene. Elements associated with de novo cardiac genes were removed if they were also associated to an ASM or FGF-MAPK activated at 18 hpf gene.

Accessibility of motif sites in differentially accessible (FDR < 0.05) peaks in the 10 hpf model was calculated using chromVAR (version 0.99.3, *Schep et al., 2017*). Peaks were resized to 200 bp. Deviations were computed between all 10 hpf conditions as well as the 6 hpf controls. Replicates were

grouped by both conditions and time. We tested for significantly differentially accessible motifs (FDR < 0.01) using the 'differentialDeviations' function from chromVAR.

### chromVAR algorithm

The accessibility of a motif is calculated by summing the fragment counts for all accessible elements containing the motif. An expected accessibility of the motif is calculated from the total counts for a motif from all replicates normalized to the library size of each replicate. For each biological replicate, the raw accessibility deviation of each motif is defined as the difference of the observed counts and the expected counts divided by the expected counts. A normally distributed background accessibility deviation for each motif is calculated by iteratively sampling accessible elements from the same replicate. The deviation Z-score for each motif is given by the difference of the raw accessibility deviation and the mean of the background accessibility deviation divided by the standard deviation of the background accessibility deviation. For a more thorough description of the algorithm underlying chromVAR, see *Schep et al. (2017)*.

## Bulk RNA-seq library preparation, Sequencing and Analyses

We used bulk RNA-seq performed following defined perturbations of FGF-MAPK signaling and FACS (*Wang et al., 2019*). To profile transcriptomes of FACS-purified cells from $Foxf^{CRISPR}$ and control samples, 1,000 cells were directly sorted in 100 µl lysis buffer from the RNAqueous-Micro Total RNA Isolation Kit (Ambion). For each condition, samples were obtained in two biological replicates. The total RNA extraction was performed following the manufacturer's instruction. The quality and quantity of total RNA was checked using Agilent RNA Screen Tape (Agilent) using 4200 TapeStation system. RNA samples with RNA Integrity Number (RIN) > 9 were kept for downstream cDNA synthesis. 250–2000 pg of total RNA was loaded as a template for cDNA synthesis using the SMART-Seq v4 Ultra Low Input RNA Kit (Clontech) with template switching technology. RNA-Seq Libraries were prepared and barcoded using Ovation Ultralow System V2 1–16 (NuGen). Up to six barcoded samples were pooled in one lane of the flow cell and sequenced by Illumina NextSeq 500 (MidOutput run). Paired-end 75 bp length reads were obtained from all the bulk RNA-seq libraries. Bulk RNA-seq libraries were aligned using STAR 2.5.3a (*Zhang et al., 2018*) with the parameters '`-run-ThreadN 20 –bamRemoveDuplicatesType UniqueIdentical`'. Counts were obtained using htseq-count (0.6.1p1) (*Anders et al., 2015*). Differential expression was calculated using edgeR for the 10 hpf conditions and DESeq2 for the conditions at 18 hpf.

## Cell-type-specific expression gene sets

We tested for differential expression in the bulk RNA-seq data for the pairwise comparisons $Foxf^{CRISPR}$ vs. $control^{CRISPR}$, $Foxf{>}M\ Ras^{CA}$ vs. control ($Foxf{>}nls{::}LacZ$), $Hand{-}r{>}Fgfr^{DN}$ vs. control ($Hand{-}r{>}nls{::}LacZ$), and $Foxf{>}M\ Ras^{CA}$ vs. $Hand{-}r{>}Fgfr^{DN}$ using DESeq2 1.16.1 (*Love et al., 2014*). We defined genes downregulated in $Foxf^{CRISPR}$ vs. $control^{CRISPR}$ ($\log_2$(FC) < −0.75 and FDR < 0.05) as Foxf targets. We defined 'MAPK-inhibited genes at 18 hpf' as the intersection of genes downregulated in $Foxf{>}M\ Ras^{CA}$ vs. control and genes downregulated in $Foxf{>}M\ Ras^{CA}$ vs. $Hand{-}r{>}Fgfr^{DN}$ ($\log_2$(FC) < −1 and FDR < 0.05) as. We defined the intersect of genes downregulated in $Hand{-}r{>}Fgfr^{DN}$ vs. control ($\log_2$(FC) < −1 and FDR < 0.05) and genes upregulated in $Foxf{>}M\ Ras^{CA}$ vs. $Hand{-}r{>}Fgfr^{DN}$ ($\log_2$(FC) > 1 and FDR < 0.05) as 'MAPK activated genes' (*Figure 4—figure supplement 1A–D*). For *Figure 4B* and *Figure 4—figure supplement 3A*, we defined the union of differentially expressed genes in either $Foxf{>}M\ Ras^{CA}$ vs. control or $Hand{-}r{>}Fgfr^{DN}$ vs. control as 'differentially expressed upon FGF-MAPK perturbation at 18 hpf.'

We integrated microarray data to obtain gene sets for MAPK perturbation at 10 hpf (*Christiaen et al., 2008*). We defined genes downregulated in $Mesp{>}Fgfr^{DN}$ vs. control ($\log_2$(FC) < −1 and $p$ < 0.05) as 'MAPK activated genes at 10 hpf.' We defined genes upregulated in $Mesp{>}Fgfr^{DN}$ vs. control ($Mesp{>}nls{::}LacZ$) ($\log_2$(FC) > 1 and $p$ < 0.05) as 'MAPK inhibited genes at 10 hpf.'

From this same data set we obtained genes downregulated in control 6 hpf vs. 10 hpf ($\log_2$(FC) < −1 and $p$ < 0.05) and upregulated in $LacZ$ 6 hpf vs. 10 hpf ($\log_2$(FC) > 1 and $p$ < 0.05).

## Statistics

We used R (version 3.6.1) (*Rizzo, 2016*) to perform all statistical analysis. An alpha level of 0.05 was adapted for statistical significance throughout the analyses. We corrected for multiple hypothesis testing using false discovery rate (FDR) where indicated. All other *p*-values are unadjusted. For statistics presented in FISH panels, a Fisher's exact test was run for each condition vs. the control. Embryos were classified as wild type, reduced expression, or no expression. A *p*-value less than 0.01 was considered significant (*Figure 2F*; *Figure 2—figure supplement 5*; *Figure 3F*; *Figure 4G*; *Figure 5—figure supplement 1B,E*). The two-tailed binomial tests in *Figure 1—figure supplements 1C*, *2D* and *3G* assume accessibility to not be feature-specific and dependent only on the fraction of the genome in base pairs covered by a feature. If promoter sites comprise 10% of the genome, we expect 10% of accessible elements will overlap with promoter sites by chance. For *Figure 3—figure supplement 7A and B*, we performed a two-tailed binomial test on the joint probability of each intersection against the null hypothesis that the probability of each set is independent. The null probability was calculated as the product of the probabilities of all constituent sets of an intersection. This is similar to the method used to calculate the deviation score in the original UpSet plot (*Lex et al., 2014*).

## Software

All computations were performed on an x86_64-centos-linux-gnu platform. In addition to software specified elsewhere in the Materials and methods section, we created a SQLite database using RSQLite 2.1.1 (*Müller et al., 2018*), ComplexHeatmap (2.0.0) (*Gu et al., 2016*), circlize (0.4.6) (*Gu et al., 2014*), latticeExtra (0.6–28) (*Sarkar and Andrews, 2016*) and Integrative Genomics Viewer (*Robinson et al., 2011*).

## Code availability

All analyses were done with publicly available software. All scripts for data analysis are available at https://github.com/ChristiaenLab/ATAC-seq. (*Wiechecki and Racioppi, 2019*; https://github.com/elifesciences-publications/ATAC-seq).

## Data availability

All sequencing data were deposited on GEO with accession GSE126691.

# Acknowledgements

We are grateful to Karen Lam, Sana Badri, Emily Miraldi and Richard Bonneau for help processing ATAC-seq data in the early phase of this study. We thank Dayanne M Castro for help and discussion on ATAC-seq data analysis. We thank Emily Huang for TVC-specific enhancers characterization, Tina Jiang for *Tbx1/10* enhancer validation, Elizabeth Zelid and Carly Vaccaro for help with reporter assays. We thank Alberto Stolfi for assistance on cloning and constructive input and discussions. We are grateful to Wei Wang for sharing single cell and bulk RNA-seq data as well as Florian Razy-Krajka for sharing $Mek^{S216D,S220E}$ prior to publication and to Pui Leng Ip for support with the FACS. We thank Esteban Mazzoni and members of the Christiaen lab for discussions, in particular Yelena Bernadskaya and Keaton Schuster for comments on the manuscript and figures. We thank Tatjana Sauka-Spengler for advice with ATAC-seq protocol. CR has been supported by a long-term fellowship ALTF 1608–2014 from EMBO. This work was funded by awards R01 HL108643 from NIH/NHLBI, R01 HD096770 from NIH/NICHD, and 15CVD01 from the Leducq Foundation to LC.

# Additional information

## Funding

| Funder | Grant reference number | Author |
| --- | --- | --- |
| European Molecular Biology Organization | Long Term Fellowship | Claudia Racioppi |

| National Heart, Lung, and Blood Institute | R01 HL108643 | Lionel Christiaen |
| Eunice Kennedy Shriver National Institute of Child Health and Human Development | R01 HD096770 | Lionel Christiaen |
| Fondation Leducq | 15CVD01 | Lionel Christiaen |

The funders had no role in study design, data collection and interpretation, or the decision to submit the work for publication.

**Author contributions**
Claudia Racioppi, Conceptualization, Data curation, Formal analysis, Funding acquisition, Validation, Investigation, Visualization, Writing—original draft, Writing—review and editing; Keira A Wiechecki, Data curation, Software, Formal analysis, Validation, Investigation, Visualization, Writing—original draft, Writing—review and editing; Lionel Christiaen, Conceptualization, Resources, Supervision, Funding acquisition, Methodology, Writing—original draft, Project administration, Writing—review and editing

**Author ORCIDs**
Claudia Racioppi ⓘ https://orcid.org/0000-0001-8117-1124
Keira A Wiechecki ⓘ https://orcid.org/0000-0003-0572-6284
Lionel Christiaen ⓘ https://orcid.org/0000-0001-5930-5667

**Decision letter and Author response**
Decision letter https://doi.org/10.7554/eLife.49921.sa1
Author response https://doi.org/10.7554/eLife.49921.sa2

# Additional files

**Supplementary files**
• Supplementary file 1. Description of ATAC-seq dataset, with sample name, conditions, type of cells, number of mapped reads for all the biological replicates.

• Supplementary file 2. Differentially FGF-MAPK-regulated genes (microarray) associated with differentially accessible peaks (ATAC-seq) in the pairwise comparison $Mesp>Fgfr^{DN}$ vs. $Mesp>LacZ$ – 10 hpf; genomic coordinates of each peak are reported.

• Supplementary file 3. Known enhancers for TVC-specific genes characterized in previous studies; genomic coordinates of each peak are reported.

• Supplementary file 4. Primer sequences used for cloning putative regulatory regions for functional enhancer assay.

• Supplementary file 5. Primer sequences of single guide RNAs used to induce CRISPR/Cas9-mediated deletions in ATAC-seq peak as well as in $Foxf$ ($Foxf^{CRISPR}$) and to induce point mutations in $Tbx1/10$ STVC-specific enhancer; genomic coordinates and peak IDs are reported for each peak validated.

• Supplementary file 6. ATM gene selected based on expression data deposited in ANISEED (https://www.aniseed.cnrs.fr).

• Supplementary file 7. Differentially expressed genes from RNA-seq on FACS-purified cells following CRISPR/Cas9-induced loss of function of $Foxf$.

• Supplementary file 8. Differentially accessible peaks from ATAC-seq on FACS-purified cells following CRISPR/Cas9-induced loss of function of $Foxf$ ($Foxf^{CRISPR}$ vs. $Control^{CRISPR}$ at 10 hpf). Sheet1: closed in $Foxf^{CRISPR}$ $\log_2(FC) < -0.5$ and FDR < 0.05; sheet2: open in $Foxf^{CRISPR}$ $\log_2(FC) > 0.5$ and FDR < 0.05.

• Supplementary file 9. Differentially accessible peaks from ATAC-seq, closed in the founder cells ($LacZ$ 6 hpf < $LacZ$ 10 hpf) and closed in $Foxf^{CRISPR}$ vs. $Control^{CRISPR}$ at 10 hpf.

- Supplementary file 10. Differentially accessible peaks closed in *Fgfr^DN* at 10 hpf and associated with de novo cardiac (sheet 1) and pharyngeal muscle (sheet 2) expressed genes.

- Supplementary file 11. Differentially FGF-MAPK-regulated genes associated with differentially accessible peaks in MAPK activated −18 hpf (sheet1) and MAPK inhibited −18 hpf (sheet 2); genomic coordinates of each peak are reported.

- Supplementary file 12. FGF-MAPK-regulated genes associated with ATAC-seq peaks that are non-differentially accessible in MAPK activated −18 hpf and MAPK inhibited −18 hpf.

- Supplementary file 13. Accessible elements associated with de novo expressed heart and pharyngeal muscle markers into pre-accessible/primed or de novo accessible elements; genomic coordinates of each peak are reported.

- Supplementary file 14. FASTA sequences including the Foxf proteins (with their accession numbers) used for *Figure 3—figure supplement 2*.

- Transparent reporting form

## Data availability

All sequencing data were deposited on GEO with accession GSE126691.

The following dataset was generated:

| Author(s) | Year | Dataset title | Dataset URL | Database and Identifier |
|---|---|---|---|---|
| Racioppi C, Wiechecki K, Christiaen L | 2019 | Genome-wide maps of chromatin accessibility governing the transition from founder cells to distinct fate-restricted cardiopharyngeal precursors | https://www.ncbi.nlm.nih.gov/geo/query/acc.cgi?acc=GSE126691 | NCBI Gene Expression Omnibus, GSE126691 |

The following previously published datasets were used:

| Author(s) | Year | Dataset title | Dataset URL | Database and Identifier |
|---|---|---|---|---|
| Wang W, Niu X, Stuart T, Jullian E, Mauck WM, Kelly RG, Satija R, Christiaen L | 2019 | A single cell transcriptional roadmap for cardiopharyngeal fate diversification | https://www.ncbi.nlm.nih.gov/geo/query/acc.cgi?acc=GSE99846 | NCBI Gene Expression Omnibus, GSE99846 |
| Christiaen L, Davidson B, Kawashima T, Powell W, Nolla H, Vranizan K, Levine M | 2008 | Transcription profiling of B7.5 lineage cells in the tunicate Ciona intestinalis to study cardiac cell migration | https://www.ebi.ac.uk/arrayexpress/experiments/E-MEXP-1478/ | ArrayExpress, E-MEXP-1478 |
| Razy-Krajka F, Lam K, Wang W, Stolfi A, Joly M, Bonneau R, Christiaen L | 2014 | Collier/OLF/EBF-dependent transcriptional dynamics control muscle specification from cardiopharyngeal progenitors | https://www.ncbi.nlm.nih.gov/geo/query/acc.cgi?acc=GSE54746 | NCBI Gene Expression Omnibus, GSE54746 |

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
