## [Decision Letter]

**Acceptance summary:**

This article provides fundamental insights regarding regulatory mechanisms that dictate precise gene expression and thus ensure proper cell fate specification. By leveraging an exceptionally well-characterized heart and pharyngeal muscle lineage gene network in an invertebrate chordate model, this study provides profound insights into principles governing reliable and accurate network deployment. The authors employ in-depth, comprehensive chromatin accessibility profiling and extensive functional manipulations to generate substantive support for a novel "combined enhancer" model. According to this model, precise regulation of crucial transcription factors is mediated by heterologous combinations of regulatory elements with distinct inputs and chromatin accessibility profiles. Strikingly, these heterologous elements function synergistically as a regulatory unit to generate a single, precise spatio-temporal expression pattern. The authors convincingly demonstrate that removal of individual elements within these combinatorial units or deployment of homologous combinations disrupt expression and can lead to improper specification. Should future studies indicate that such combinatorial regulatory elements are broadly deployed, this would have wide-ranging implications regarding gene network architecture as well as the potential impact of non-coding mutations in the context of evolution and disease.

**Decision letter after peer review:**

Thank you for submitting your article "Combinatorial chromatin dynamics foster accurate cardiopharyngeal fate choices" for consideration by *eLife*. Your article has been reviewed by three peer reviewers, and the evaluation has been overseen by a Reviewing Editor and Didier Stainier as the Senior Editor. The reviewers have opted to remain anonymous.

The reviewers have discussed the reviews with one another and the Reviewing Editor has drafted this decision to help you prepare a revised submission.

There was a strong consensus among the reviewers that this paper contains findings that are of broad general interest and are appropriate for publication in *eLife* once revisions have been made to more clearly align their data with their conclusions. In particular, the reviewers were impressed by the broad scope of their ATAC-seq analysis, the innovative application of this technique to Ciona, some of the more in-depth findings regarding specific regulatory elements and some of the insights provided by this analysis in regards to more general principles of regulatory logic.

However, the reviewers had a number of critical concerns including a lack of clarity in some key definitions and figures, some questions regarding interpretation of specific assays and experiments and a lack of sufficient support for some of their major conclusions including the proposed combined enhancer model.

The was also a consensus that these concerns can be addressed in two ways. The authors can either modify their conclusions to better match their current data or they can conduct further experiments to more robustly support their current conclusions.

Reviewer #1:

In this study, Racioppi, Wiechecki and Christiaen have deployed an impressive series of genome-wide analyses complemented with CRISPR/Cas9-mediated genome editing. The study addresses chromatin accessibility profiles during early development of cardiopharyngeal lineages in Ciona. In this lineage, three rounds of asymmetric cell divisions, which are controlled by differential activation of FGF-MAPK signals in a reiterative manner, lead to generation of four cell types, first heart precursor (FHP), second heart precursor (SHP), atrial siphon muscle precursor (ASMP) and anterior tail muscle (ATM). The Christiaen laboratory is one of the major players in deciphering the underlying transcriptional and signalling control of the cardiopharyngeal lineage segregation. The current study reveals another layer of such regulatory mechanisms, in particular, chromatin accessibility, by deploying ATAC-seq on isolated cardiopharyngeal lineage cells. The study contains a huge amount of bioinformatic analyses, which, to be honest, are very hard for me to digest. Nonetheless, I really appreciate the last part of their study highlighted in Figures 5, Figure 5—figure supplements 1 and 2, which focuses on chromatin accessibility profiles of two key genes of cardiopharyngeal fate choices, *Ebf* and *Tbx1/10*. It reveals convincingly that these genes are controlled by multiple *cis*-regulatory elements with distinct temporal chromatin accessibility profiles and distinct TF-binding motif compositions. This finding indeed represents the main message of the current study.

I have a few comments (questions):

1) For Figure 5A and Figure 5—figure supplement 2A, it would be informative to include the ATAC-seq dataset of FoxF(CRISPR)-10 hpf.

2) I assume that changes in chromatin accessibility between FGFR(DN)-10 hpf and FoxF(CRISPR)-10 hpf would exhibit a certain degree of correlation since FoxF is transcriptionally activated by FGF signals in TVCs. It would be nice if the authors could conduct a correlation test. There seems to be such a correlation for the peak ID *KhL20.169* within *Gata4/5/6* locus (Figure 3—figure supplement 2A and Figure 3—figure supplement 4D).

3) Why do the authors call MyoD905>GFP cells as mesenchymal cells? Aren't they muscle cells (Christiaen et al., 2008)?

4) The authors mention "cardiopharyngeal markers" that include TVC progenitor markers, primed cardiac markers, primed ASM markers, de novo cardiac markers, de novo ASM markers and ATM markers. The definition for these different markers is supposed to be described in supplementary text. I imagine that the authors are meaning that in the "Gene Set Enrichment Analysis (GSEA)" section but it doesn't help me to understand the definition. A clearer definition, together with gene lists for the different categories of cardiopharyngeal markers as a supplementary table, would be helpful.

Reviewer #2:

This manuscript by Racioppi et al. performs the first ATAC-seq experiments in the early heart cell lineages of Ciona in order to find enhancers that are accessible in sublineages of the heart. This paper creates a very valuable resource for the heart and ciona communities. I don't feel that the authors make the most of their data and the value of this work is skipped over to focus on the combined enhancer. Overall I find there are too many interpretations/extrapolations without sufficient evidence and these detract from the excellent quality and substantial value of this dataset for understanding heart development. I would recommend that the authors focus on their data and what it tells us about the regulatory networks and developmental programs of the heart rather than the concept of combined enhancers.

Combined enhancers:

A major claim of the paper, that "combined enhancers" foster spatially and temporally accurate fate choices, by increasing the repertoire of regulatory inputs that control gene expression, through either accessibility and/or activity" is only supported by a study of only one locus shown in Figure 6, and I have issues with how the experiment is designed. The authors do an experiment in Figure 6 where they multimerize a weak enhancer (1x, 2x, 3x) and find that when 3 copies of the weak enhancer are put next to each other in a certain way, they find ectopic expression by reporter assay. They then say that this, coupled with evidence that many weak enhancers in the genomic context do not cause ectopic expression, show that these "combined enhancers" foster spatially and temporally accurate fate choices. However, this is comparing apples (3 identical enhancers with non-endogenous spacing together on a plasmid reporter) to oranges (3 unique endogenous enhancers with endogenous spacing in the genome). Since spacing between enhancers and genomic context can affect transcriptional output, I would believe this more if they put the 3 unique weak enhancers from the genomic context into a synthetic construct with spacings like the 3x multimerized enhancer and showed that these constructs did not yield ectopic expression. Even if this experiment were to show that the different inputs led to specific expression, I don't understand how this type of enhancer is different from enhancers using different inputs turning on at different times and space during a developmental program.

Definition of Shadow enhancer: The authors state that "shadow enhancer promotes robust transcription through the actions of multiple elements mediating similar regulatory inputs". Based on the literature I wasn't able to find a consensus that shadow enhancers have to use similar inputs, indeed it is very hard to pin down exactly what the inputs for many shadow enhancers are. Perry, 2011/Hong, 2008.

Reviewer #3:

In this manuscript, Racioppi et al. investigate the temporal dynamics of chromatin accessibility during the different steps of cardiopharyngeal progenitor specification in larvae of the chordate Ciona. They sample different cell types at different time points post fertilization, ranging from *Mesp1+* founder cells after 6 hours to first and second heart progenitors, as well as atrial syphon precursors and anterior tail muscle cells after 18 hours. They show that global accessibility decreases with time, and that most enhancers are opened early during the process, in multipotent progenitors, even though their associated genes are mostly activated later. Using CRISPR-mediated deletions and reporter assays, they demonstrate that the enhancers they identify are necessary and sufficient to activate gene expression with temporal and spatial specificity. They then identify putative regulators of stage- and cell-type specific enhancer accessibility and confirm *Foxf* as a driver of enhancer accessibility in the cardiac lineage. Then, they study the locus of *Ebf*, where multiple enhancers are necessary for its expression in atrial muscle precursors. Finally, they study combinatorial effects of *Ebf* enhancers using a reporter assay and find that multiple copies of the same enhancer increase transcription efficiency, at the cost of precocious activation, whereas one copy of a combined enhancer construct recapitulates gene expression with greater temporal fidelity.

In this well-conducted study, the authors present solid data to back their conclusions, which are interesting for the field of development. Some points could be addressed to strengthen their claims:

- Does their reporter assay recapitulate enhancer accessibility or activity? This is unclear because the loss of the genomic context may cause an enhancer to be more easily activated. What is the timing of activation of the reporter gene in regard to the timing of chromatin accessibility and gene transcription activation?

- For the enhancers they define as being dependent on *Foxf*, does the knockout of *Foxf* impede their activation in a reporter assay? Or are enhancers containing mutated *Foxf* sites inactive (endogenously or in a reporter essay)?

- The authors could use histone marks to assess if the decoupling between accessibility and gene activation is also seen between gene activation and accumulation of histone marks typical of active genetic elements, such as histone 3 lysine 27 acetylation, if it is technically possible for them with their low number of cells.

- Figure 4A-C and the corresponding text are unclear. It is difficult to understand how the later comparison of FGF-MAPK perturbation relates to the rest of the paper.

---

## [Author Response]

Reviewer #1:[…] I have a few comments (questions):1) For Figure 5A and Figure 5—figure supplement 2A, it would be informative to include the ATAC-seq dataset of FoxF(CRISPR)-10 hpf.

We have included Foxf^CRISPR^ ATAC-seq data at 10 hpf in Figure 5B, and in Figure 5—figure supplement 2A.

2) I assume that changes in chromatin accessibility between FGFR(DN)-10 hpf and FoxF(CRISPR)-10 hpf would exhibit a certain degree of correlation since FoxF is transcriptionally activated by FGF signals in TVCs. It would be nice if the authors could conduct a correlation test. There seems to be such a correlation for the peak ID *KhL20.169* within *Gata4/5/6* locus (Figure 3—figure supplement 2A and Figure 3—figure supplement 4D).

This is correct. We observed several peaks showing similar change in chromatin accessibility following FGF perturbation (Fgfr^DN^) or tissue-specific *Foxf* loss-of-function (Foxf^CRISPR^), like peaks *KhC14.806,.807* and.*808* associated to *Hand* in Figure 3E, peaks *KhC4.137* and.*144* in the *Lrp4/8* locus in Figure 4D, and peaks *KhC1.479* in the *Eph1* locus, *KhC9.3113* in the *Ddr1/2* locus, *KhC4.606* in the *Smurf1/2* locus, *KhC6.1052* and *KhC6.1046* in the *Fzd4* locus shown in Figure 3—figure supplement 4.

Consistent with this observation, we found that chromatin accessibility changes induced by Mesp>Fgfr^DN^ and Foxf^CRISPR^ compared to controlsat 10 hpf are correlated, with a Spearman correlation coefficient (𝜌) of 0.49. We added this additional information in Figure 3—figure supplement 2F as a scatter plot to compare the correlation of differentially accessible regions in “Mesp>Fgfr^DN^ and “Foxf^CRISPR^ at 10 hpf” ATAC-seq samples.

3) Why do the authors call MyoD905>GFP cells as mesenchymal cells? Aren't they muscle cells (Christiaen et al., 2008)?

The *cis*-regulatory DNA from the myogenic differentiation factor *MyoD* (termed *Myod905*) was used to express green fluorescent protein in the mesenchymal cells (Christiaen at al., 2008). This is because the enhancer does not contain all control elements of Myod/Mrf and “leaks” strongly in the mesenchyme. This enhancer is active in the tail muscle cells as well. Nevertheless, mesenchymal cells are smaller and more numerous than tail muscle cells, they are thus more efficiently isolated by FACS than tail muscle cells. There is a possibility that muscle cells are sorted along with mesenchymal cells but they will be underrepresented in the FACSorted cell population. In Christiaen et al., 2008, the Myod905>YFP^Venus^+ population expressed higher levels of mesenchyme markers.

In Christiaen et al., 2008 (Supplementary Figure 1), we described a similar trend for the two sister cells, TVC and ATM. TVC cells are smaller than ATMs and they are more efficiently sorted by FACS rather than their sister cells ATMs (Christiaen et al., 2008, Supporting material section 3.2).

In order to be more accurate in defining ATAC-seq samples isolated from mesenchymal cells at different time points, we replaced “Mesenchyme” ATAC-seq samples with “MyoD905>GFP”.

*4) The authors mention "cardiopharyngeal markers" that include TVC progenitor markers, primed cardiac markers, primed ASM markers,* de novo *cardiac markers,* de novo *ASM markers and ATM markers. The definition for these different markers is supposed to be described in Supplementary text. I imagine that the authors are meaning that in the "Gene Set Enrichment Analysis (GSEA)" section but it doesn't help me to understand the definition. A clearer definition, together with gene lists for the different categories of cardiopharyngeal markers as a supplementary table, would be helpful.*

In the Materials and methods under the section “Definition of gene sets”, we clarified the definition of gene sets used for "Gene Set Enrichment Analysis (GSEA)" by listing the cell-type-specific gene sets including primed and de novo-expressed Cardiac and ASM/pharyngeal muscle, as defined from scRNA-seq by Wang et al., 2019, as well as anterior tail muscle (ATM) gene sets.

More specifically, we replaced:

“Gene Set Enrichment Analysis (GSEA)

We define gene sets as described in the supplementary text. […] We measured enrichment for peaks associated to the gene categories defined by both bulk RNA-seq and scRNA-seq (Wang et al., 2019) in open peaks (indicated by a positive enrichment score, ES) or closed peaks (indicated by a negative ES) in each comparison.”

With:

“Definition of gene sets

We used cardiopharyngeal lineage cell-type-specific gene sets (cardiopharyngeal markers), including primed and de novo-expressed Cardiac and ASM/pharyngeal muscle as well as secondary TVC (STVC), first heart precursor (FHP), second heart precursor (SHP) markers defined by scRNA-seq (Wang et al., 2019). […] To these we added the gene sets either activated or inhibited by MAPK at 10 or 18 hpf as defined by microarray (Christiaen et al., 2008) and Bulk RNA-seq analysis (Wang et al., 2019).”

“Gene Set Enrichment Analysis (GSEA)

We performed GSEA on elements associated to these gene sets using fgsea 1.2.1 (Sergushichev, 2016) with parameters: minSize=5, maxSize=Inf, nperm=10000. Because the gene sets used for GSEA need not be exclusive, we can use ATAC-seq peaks as “genes” and define a gene set as all peaks annotated to genes in the set. We measured enrichment for peaks associated to the gene sets in open peaks (indicated by a positive enrichment score, ES) or closed peaks (indicated by a negative ES) in each comparison.”

Furthermore, we added “We performed bulk RNA-seq following defined perturbations of FGF-MAPK signaling and FACS (Wang et al., 2019).” as the first sentence under Bulk RNA-seq Library Preparation, Sequencing and Analyses.

We deleted:

“To run GSEA, we used gene sets including primed and de novo-expressed Cardiac and ASM/pharyngeal muscle, as well as secondary TVC (STVC), first heart precursor (FHP), second heart precursor (SHP) markers defined by scRNA-seq(Wang et al., 2019). Anterior tail muscles (ATM) marker genes were derived from publicly available expression data (Brozovic et al., 2018) (Tables S46). The gene sets either activated or inhibited by MAPK at 10 or 18 hpf were derived from bulk RNA-seq following defined perturbations of FGF-MAPK signaling and FACS (Wang et al., 2019).”

Finally, we replaced “Supplementary text” with “Materials and methods”.

Reviewer #2:This manuscript by Racioppi et al. performs the first ATAC-seq experiments in the early heart cell lineages of Ciona in order to find enhancers that are accessible in sublineages of the heart. This paper creates a very valuable resource for the heart and ciona communities. I don't feel that the authors make the most of their data and the value of this work is skipped over to focus on the combined enhancer.

We appreciate this comment and indeed tried to not underplay the findings that pertain to the general description of accessibility patterns, and the roles of FGF-MAPK signaling and *Foxf* in establishing these patterns, as well as defining novel enhancers. Figures 1 to 4 are focused on this, and hopefully do it justice. Arguably, this dataset contains more information to be mined and followed-up on.

Overall I find there are too many interpretations/extrapolations without sufficient evidence and these detract from the excellent quality and substantial value of this dataset for understanding heart development. I would recommend that the authors focus on their data and what it tells us about the regulatory networks and developmental programs of the heart rather than the concept of combined enhancers.

Our initial rationale was that the general concepts of decoupling early accessibility from late enhancer activity, of a forkhead factor acting as a possible pioneer had become classic concepts in the past few years. We also wanted to circle back to the regulation of actual gene expression, especially heart vs. pharyngeal muscle-specific gene expression, and do justice to the novel observations that genes like *Ebf* and *Tbx1/10* were flanked by elements that showed different dynamics of accessibility. This seemed to offer an opportunity for more novel and yet broadly relevant findings, which we report as “combined enhancers”.

We added new experimental data (Figures 6 and 7), which we think strengthen our conclusions: indeed, we could experimentally decouple the level of expression of *Ebf* from its spatio-temporal accuracy and thus cause ectopic activation of the endogenous locus and fate transformation of the second heart lineage into pharyngeal muscles. These new results provide more direct experimental support for the claim that “combined enhancers foster accurate fate choices” to paraphrase the title.

We considered the option of splitting the paper, but practical constraints and the rationale explained above led us to keep the paper largely as is, while strengthening conclusions.

Combined enhancers:A major claim of the paper, that "combined enhancers" foster spatially and temporally accurate fate choices, by increasing the repertoire of regulatory inputs that control gene expression, through either accessibility and/or activity" is only supported by a study of only one locus shown in Figure 6.

We reported a similar pattern of variable dynamics of accessibility for enhancer elements for both *Tbx1/10* and *Ebf*, but could only perform the more extensive analysis on *Ebf* indeed. This part of the paper stands as a proof of a rather novel concept, and it is conceivable that future studies will be needed to probe its generality. We suspect that it will matter particularly for developmental control genes like *Ebf*, which mediate commitment and rely on autoregulation for long-term maintenance.

I have issues with how the experiment is designed. The authors do an experiment in Figure 6 where they multimerize a weak enhancer (1x, 2x, 3x) and find that when 3 copies of the weak enhancer are put next to each other in a certain way, they find ectopic expression by reporter assay. They then say that this, coupled with evidence that many weak enhancers in the genomic context do not cause ectopic expression, show that these "combined enhancers" foster spatially and temporally accurate fate choices. However, this is comparing apples (3 identical enhancers with non-endogenous spacing together on a plasmid reporter) to oranges (3 unique endogenous enhancers with endogenous spacing in the genome). Since spacing between enhancers and genomic context can affect transcriptional output, I would believe this more if they put the 3 unique weak enhancers from the genomic context into a synthetic construct with spacings like the 3x multimerized enhancer and showed that these constructs did not yield ectopic expression.

We extensively addressed this point experimentally and basically showed that the concatemer of distinct elements without endogenous spacer sequences behave like the full length upstream sequence, in that it drives accurate expression in the pharyngeal muscles, whereas the multimer of one distal weak enhancer causes ectopic activation, and ultimately fate transformation.

We added the following paragraph in the Results section: “To test whether spacing between accessible elements could affect transcriptional output, we built a concatemer of *KhL24.37,.36,.35*, and.*34* elements without endogenous spacer sequences. This construct increased the proportion of embryos with ASM cells expressing the reporter to 92% ± 2% (SE, n=130; Figure 6C), but it did not induce ectopic expression in the second heart lineage”, and: “neither one copy of *KhL24.37* (1x *KhL24.37*), nor the full combined enhancers, with or without endogenous spacers, sufficed to cause substantial fate transformation of *Tbx1/10*+ second heart lineage into migratory pharyngeal muscle precursors (Figure 7C-D)”.

Even if this experiment were to show that the different inputs led to specific expression, I don't understand how this type of enhancer is different from enhancers using different inputs turning on at different times and space during a developmental program.

In this case, the key difference is that these elements contribute to the onset of *Ebf* expression for ~ 1 hour in the pharyngeal muscle precursors (as we showed in Razy-Krajka et al., 2018). In other words, their actions are at the exact same time and place in development. This is thus different from classic modular *cis*-regulatory system paradigms such as *Drosophila* Even-skipped, and many other examples.

Definition of Shadow enhancer: The authors state that " shadow enhancer promotes robust transcription through the actions of multiple elements mediating similar regulatory inputs" Based on the literature I wasn't able to find a consensus that shadow enhancers have to use similar inputs, indeed it is very hard to pin down exactly what the inputs for many shadow enhancers are. Perry, 2011/Hong, 2008.

Three main features characterized shadow enhancers: (1) they each drive a pattern of transcription resembling that of a previously identified “primary” enhancer that is more proximal to the promoter being regulated; (2) they bind the same TFs as the primary enhancer, suggesting a similar regulatory logic; and (3) they are located either within an intron of, or on the far side of a neighboring gene (Barolo et al., 2012).

Based on the literature, Mike Levine’s lab was the first to coin the term “Shadow Enhancer” for “remote secondary enhancers mapping far from the target gene and mediating activities overlapping the primary enhancer”. This paper is cited in the Introduction (Hong et al., 2008).

Moreover, we cited the “Zeitlinger et al., 2007” paper where Levine’s group described the identification of shadow enhancers by chromatin immunoprecipitation (ChIP)-chip assays.

The corresponding author was still a postdoc in the Levine lab when these studies were conducted and is thus familiar with the issue. Basically the concept of shadow enhancer focuses primarily on the transcriptional output, and the fact that shadow enhancers integrate the same inputs is generally implied, and was an important part of their discovery.

More recently, studies are beginning to uncover that shadow enhancers integrate distinct inputs. This is essentially what we are saying for combined enhancers, except that we show differences in the dynamics of chromatin accessibility, which is particularly innovative here.

Reviewer #3:[…] In this well-conducted study, the authors present solid data to back their conclusions, which are interesting for the field of development. Some points could be addressed to strengthen their claims:- Does their reporter assay recapitulate enhancer accessibility or activity? This is unclear because the loss of the genomic context may cause an enhancer to be more easily activated.

For several elements analyzed in this study, we observed that reporter gene assays recapitulate enhancer activity and not accessibility. We added additional information to support this point on Figure 4—figure supplement 2.

In the third paragraph of the subsection “Chromatin accessibility in late heart vs. pharyngeal muscle precursors”, we described that very few de novo-expressed pan-cardiac genes (30) were also differentially expressed upon FGF-MAPK perturbation at 18 hpf and that only 8 out of these 30 genes were associated with differentially accessible element following perturbation of FGF-MAPK signaling (Figure 4A and B). In this group of 8 elements, we identified the *KhC5.1641* peak that is hosted in the 5’UTR of the de novo expressed cardiac marker, *Mmp21*, gene locus (Figure 4—figure supplement 2C).

We experimentally tested a ~3kb region upstream the coding ATG of *Mmp21* encompassing the *KhC5.1641* peak by gene reporter assay (Figure 4—figure supplement 2C). Even if *Mmp21* is expressed only in the first heart precursors (Wang et al., 2009), the ~3kb region was able to activate GFP in both cardiac (first and second heart precursors) and pharyngeal muscle cell precursors in embryos at stage 25 embryos (18hpf) (Figure 4—figure supplement 2D and E).

Similarly, we identified an element (peak ID: *KhC2.3468*) upstream the coding region of the de novo expressed pharyngeal muscle marker, *Tmct2*. This peak was exclusively accessible in the pharyngeal cell upon perturbation of FGF-MAPK signaling (Figure 4—figure supplement 2F). However, by reporter assay, it was able to activate GFP expression in the whole B7.5 cell lineage, including ATMs and cardiac and pharyngeal cell muscles (Figure 4—figure supplement 2G).

We additionally add another example regarding two accessible elements upstream the coding region of the primed-expressed cardiac marker, *KH.C1.1093_ZAN* that were able to activate GFP in both cardiac and pharyngeal cell muscles (Figure 4—figure supplement 2H, I).

We added the following paragraph in the Results section:

“Similarly, reporter gene expression assays showed that DNA fragments containing differentially accessible elements located ~0.5 kb upstream of the coding region of the de novo-expressed gene *Tmtc2 (KhC2.3468*), and upstream of *KH.C1.1093_ZAN (KhC3.47, KhC3.46*), were sufficient to drive GFP expression in both cardiac and pharyngeal muscle progenitors, consistent with the notion that electroporated plasmids are not “chromatinized” and thus constitutively accessible (Figure 4—figure supplement 2F-G; Figure 4—figure supplement 3C-D). This suggested that, for genes like *Mmp21, Tmtc2* and *Zan*, cell-type-specific accessibility determines cardiac vs. pharyngeal muscle-specific gene expression.”

What is the timing of activation of the reporter gene in regard to the timing of chromatin accessibility and gene transcription activation?

We observed different timing of the reporter gene activation in regard to the timing of chromatin accessibility and gene transcription. We often observed a decoupling between chromatin accessibility and chromatin activity for de novo expressed genes. For example, the previously characterized STVC-specific enhancer located upstream the starting coding of *Tbx1/10* (Razy-Krajka et al., 2018) was already accessible in the naive *Mesp+* mesoderm at 6 hpf preceding the gene transcription which starts at 15hpf (confirmed by in situ and RNA-seq data) (Figure 5—figure supplement 2A). We describe this in the Results section: “*Tbx1/10* showed a similar logic, whereby a constitutively accessible upstream element (*KhC7.909*) acts as an enhancer of cardiopharyngeal expression (Razy-Krajka et al., 2018)”. Similarly, the TVC-specific enhancer of *Foxf* (Beh et al., 2007) is accessible in founder cells at 6 hpf anticipating the gene expression which starts immediately after the founder cells divide in two sister cells, TVC and ATM, at 8 hpf (Figure 3—figure supplement 2A). We describe this in the Results section: “Finally, the *Foxf* enhancer was accessible in naive *Mesp+* founder cells, suggesting that it is poised for activation, unlike the intronic *Gata4/5/6* enhancer (Figure 3—figure supplement 2A).”

The pharyngeal muscle determinant *Ebf* gene locus is surrounded by multiple regions showing temporal dynamics in chromatin accessibility during cardiopharyngeal development (Figure 5A). However, the minimal enhancer driving *Ebf* in the pharyngeal muscles (*KhL24.37* peak) opens only at 15hpf, when the gene turns on in the ASMPs. We describe this in the Results section: “Both loci contained multiple accessible regions, including elements open already in the naive *Mesp*+ mesoderm (e.g. *Ebf_KhL24.35/36*), cardiopharyngeal-lineage-specific elements that open prior to gene activation but after induction of multipotent progenitors (e.g. *Ebf_KhL24.34*), and elements that open de novo in fate-restricted pharyngeal muscle precursors, where the gene is activated (e.g. *Ebf_KhL24.37*) (Figure 5A, B). Previous reporter gene expression assays identified the latter element, *Ebf_KhL24.37,* as a weak minimal enhancer with pharyngeal muscle-specific activity (Wang et al., 2013).”.

We also observed some elements opening simultaneously with the gene transcription activation. For example, TVC-specific enhancers controlling two primed-expressed genes as *GATA4/5/6* (Woznica et al., 2012) (Figure 3—figure supplement 4D) as well as *Hand* (Figure 3E) are accessible at 10 hpf when the genes are transcribed in the multipotent progenitors.

- For the enhancers they define as being dependent on Foxf, does the knockout of Foxf impede their activation in a reporter assay? Or are enhancers containing mutated Foxf sites inactive (endogenously or in a reporter essay)?

We added new data (Figure 5—figure supplement 2) to support that loss-of-function of *Foxf* in the cardiopharyngeal lineage reduce the enhancer elements that we have characterized in this work.

We identified two conserved Forkhead binding sites in the minimal STVC-specific enhancer controlling *Tbx1/10* activation (termed T12) and induced point mutation in this binding site (“mFoxf.1:TAAACA in TAACCA” and “mFoxf.2:GTTTA in GTGTA”; we listed the oligos used for point mutation in Supplementary file 5).

We tested the enhancer activity by reporter assay using T12 encompassing the two point mutations in the Forkhead binding sites and T12 wt. We found that the enhancer activity is drastically reduced in ~92% of larvae expressing T12-mFoxf.2 and ~88% of larvae expressing T12-mFoxf.1 compared to the wt T12 element (Figure 5—figure supplement 2D, E).

Moreover, in embryos where we induced loss-of-function of *Foxf* (Foxf^CRISPR^), we observed T12 enhancer activity drastically reduced compared to the CRISPR control embryos. ~84% of embryos were not expressing GFP in STVC progeny in embryos at stage 26 compared to the control embryos where ~28% of embryos were expressing GFP in both ASMPs and SHPs, accordingly to the normal activity of T12 (Figure 5——figure supplement 2F).

We added the following paragraph in the Results section: “We identified two conserved putative Forkhead binding sites in the minimal STVC-specific enhancer from the *Tbx1/10* locus (termed T12, (Razy-Krajka et al., 2018), which were necessary for reporter gene expression (Figure 5—figure supplement 2D, E). Moreover, loss-of-function of *Foxf* (Foxf^CRISPR^) drastically reduced T12 enhancer activity (Figure 5—figure supplement 2). These results are consistent with the hypothesis that *Foxf* acts directly on the minimal *Tbx1/10* enhancer to promote its activity in the second multipotent cardiopharyngeal progenitors.”

- The authors could use histone marks to assess if the decoupling between accessibility and gene activation is also seen between gene activation and accumulation of histone marks typical of active genetic elements, such as histone 3 lysine 27 acetylation, if it is technically possible for them with their low number of cells.

This would be fantastic, and our lab has been testing the recently developed Cut&Tag approach, but this is not established yet, and would extend far beyond the scope of this paper, since we carefully avoided making unsubstantiated claims about histone modifications.

- Figure 4A-C and the corresponding text are unclear. It is difficult to understand how the later comparison of FGF-MAPK perturbation relates to the rest of the paper.

In a nutshell, we observed very few differences in accessibility between fate-restricted heart and pharyngeal muscle precursors, which contrasts with differences in gene expression, and is consistent with the idea that pan-cardiopharyngeal patterns of accessibility are established early, upon induction of progenitors, and that differential activity of otherwise accessible elements is a key driver of differential gene expression, with the noted exceptions.

We added more information regarding the pan-cardiac and pharyngeal muscle genes that are differentially expressed upon FGF signaling perturbation.

We added the following paragraph in the Results section: “Similarly, out of 23 de novo-expressed pharyngeal muscle genes that were also differentially expressed upon FGF-MAPK perturbation at 18 hpf, 11 were associated with one differentially accessible element following perturbation of FGF-MAPK signaling (Figure 4—figure supplement 3A).”

These sub-set of genes are surrounded by few chromatin accessible regions showing differential accessibility in response to FGF signaling molecular perturbations. Even if few peaks were significantly changing accessibility, we identified DNA fragments located ~0.5 Kb upstream the coding region of the de novo-expressed *Tmtc2 (KhC2.3468*) and upstream of the primed-expressed *KH.C1.1093_ZAN (KhC3.47, KhC3.46*) were sufficient to drive GFP expression in both cardiac and pharyngeal muscle progenitors (Figure 4—figure supplement 2F-G; Figure 4—figure supplement 3C-D). These data along with the validation of *Lrp4/8* associated peaks in Figure 4D-G, reinforced the concept of a decoupling between enhancer activity and accessibility.